# Deep Declarative Dynamic Time Warping for End-to-End Learning of Alignment Paths

**Ming Xu**[1,2]  **Sourav Garg**[1]  **Michael Milford**[1]  **Stephen Gould**[2]
[1]Queensland University of Technology  [2]Australian National University
{mingda.xu, stephen.gould}@anu.edu.au
{s.garg, michael.milford}@qut.edu.au

## Abstract

This paper addresses learning end-to-end models for time series data that include a temporal alignment step via dynamic time warping (DTW). Existing approaches to differentiable DTW either differentiate through a fixed warping path or apply a differentiable relaxation to the min operator found in the recursive steps used to solve the DTW problem. We instead propose a DTW layer based around bi-level optimisation and deep declarative networks, which we name DecDTW. By formulating DTW as a continuous, inequality constrained optimisation problem, we can compute gradients for the solution of the optimal alignment (with respect to the underlying time series) using implicit differentiation. An interesting byproduct of this formulation is that DecDTW outputs the *optimal* warping path between two time series as opposed to a soft approximation, recoverable from Soft-DTW. We show that this property is particularly useful for applications where downstream loss functions are defined on the optimal alignment path itself. This naturally occurs, for instance, when learning to improve the accuracy of predicted alignments against ground truth alignments. We evaluate DecDTW on two such applications, namely the audio-to-score alignment task in music information retrieval and the visual place recognition task in robotics, demonstrating state-of-the-art results in both.

## 1 Introduction

The dynamic time warping (DTW) algorithm computes a discrepancy measure between two temporal sequences, which is invariant to shifting and non-linear scaling in time. Because of this desirable invariance, DTW is ubiquitous in fields that analyze temporal sequences such as speech recognition, motion capture, time series classification and bioinformatics (Kovar & Gleicher, 2003; Zhu & Shasha, 2003; Sakoe & Chiba, 1978; Bagnall et al., 2017; Petitjean et al., 2014; Needleman & Wunsch, 1970). The original formulation of DTW computes the minimum cost matching between elements of the two sequences, called an alignment (or warping) path, subject to temporal constraints imposed on the matches. For two sequences of length $m$ and $n$, this can be computed by first constructing an $m$-by-$n$ pairwise cost matrix between sequence elements and subsequently solving a dynamic program (DP) using Bellman's recursion in $O(mn)$ time. Figure 1 illustrates the mechanics of the DTW algorithm.

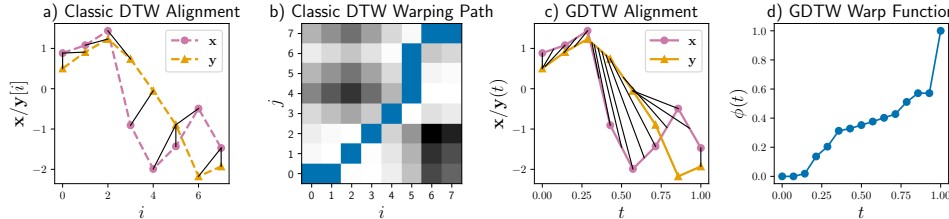

Figure 1: Classic DTW (a) is a discrete optimisation problem which finds the minimum cost warping path through a pairwise cost matrix (b). DecDTW uses a continuous time variant of classic DTW (GDTW) (c) and finds an optimal time warp *function* between two continuous time signals (d).

There has been interest in recent years around embedding DTW within deep learning models (Cuturi & Blondel, 2017; Cai et al., 2019; Lohit et al., 2019; Chang et al., 2019; 2021), with applications spread across a variety of learning tasks utilising audio and video data (Garreau et al., 2014; Dvornik et al., 2021; Haresh et al., 2021; Chang et al., 2019; 2021), especially where an *explicit* alignment step is desired. There are several distinct approaches in the literature for differentiable DTW layers: Cuturi & Blondel (2017) (Soft-DTW) and Chang et al. (2019) use a differentiable relaxation of $\min$ in the DP recursion, Cai et al. (2019) and Chang et al. (2021) observe that differentiation is possible after fixing the warping path, and others (Lohit et al., 2019; Shapira Weber et al., 2019; Grabocka & Schmidt-Thieme, 2018; Kazlauskaite et al., 2019; Abid & Zou, 2018) regress warping paths directly from sequence elements without explicitly aligning. Note, methods based on DTW differ from methods such as CTC for speech recognition (Graves & Jaitly, 2014), where word-level transcription is more important than frame-level alignment, and an explicit alignment step is not required.

We propose a novel approach to differentiable DTW, which we name DecDTW, based on deep implicit layers. By adapting a continuous time formulation of the DTW problem proposed by Deriso & Boyd (2019), called GDTW, as an inequality constrained non-linear program (NLP), we can use the deep declarative networks (DDN) framework (Gould et al., 2021) to define the forward and backward passes in our DTW layer. The forward pass involves solving for the optimal (continuous time) warping path; we achieve this using a custom dynamic programming approach similar to the original DTW algorithm. The backward pass uses the identities in Gould et al. (2021) to derive gradients using implicit differentiation through the solution computed in the forward pass.

We will show that DecDTW has benefits compared to existing approaches based on Soft-DTW (Cuturi & Blondel, 2017; Le Guen & Thome, 2019; Blondel et al., 2021); the most important of which is that DecDTW is more effective and efficient at utilising *alignment path information* in an end-to-end learning setting. This is particularly useful for applications where the objective of learning is to improve the accuracy of the alignment itself, and furthermore, ground truth alignments between time series pairs are provided. We show that using DecDTW yields both considerable performance gains and efficiency gains (during training) over Soft-DTW (Le Guen & Thome, 2019) in challenging real-world alignment tasks. An overview of our proposed DecDTW layer is illustrated in Figure 2.

**Our Contributions**    First, we propose a novel, inequality constrained NLP formulation of the DTW problem, building on the approach in Deriso & Boyd (2019). Second, we use this NLP formulation to specify our novel DecDTW layer, where gradients in the backward pass are computed implicitly as in the DDN framework (Gould et al., 2021). Third, we show how the alignment path produced by DecDTW can be used to minimise discrepancies to ground truth alignments. Last, we use our method to attain state-of-the-art performance on challenging real-world alignment tasks.

## 2    RELATED WORKS

**Differentiable DTW Layers**    Earlier approaches to learning with DTW involve for each iteration, alternating between first, computing the optimal alignment using DTW and then given the fixed alignment, optimising the underlying features input into DTW. Zhou & De la Torre (2009); Su & Wu (2019); Zhou & De la Torre (2012) analytically solve for a linear transformation of raw observations. More recent work such as DTW-Net (Cai et al., 2019) and DP-DTW (Chang et al., 2021) instead take a single gradient step at each iteration to optimise a non-linear feature extractor. All aforementioned methods are not able to directly use path information within a downstream loss function.

**Differentiable Temporal Alignment Paths**    Soft-DTW (Cuturi & Blondel, 2017) is a differentiable relaxation of the classic DTW problem, achieved by replacing the $\min$ step in the DP recursion with a differentiable *soft-min*. The Soft-DTW discrepancy is the expected alignment cost under a Gibbs distribution over alignments, induced by the pairwise cost matrix $\Delta$ and smoothing parameter $\gamma > 0$. Path information is encoded through the expected alignment matrix, $A_\gamma^* = \mathbb{E}_\gamma[A] = \nabla_\Delta \mathbf{dtw}_\gamma(\mathbf{X}, \mathbf{Y})$ (Cuturi & Blondel, 2017). While $A_\gamma^*$ is recovered during the Soft-DTW backward pass, differentiating through $A_\gamma^*$, involves computing Hessian $\nabla_\Delta^2 \mathbf{dtw}_\gamma(\mathbf{X}, \mathbf{Y})$. Le Guen & Thome (2019) proposed an efficient custom backward pass to achieve this. A loss can be specified over paths through a penalty matrix $\mathbf{\Omega}$, which for instance, can encode the error between the expected predicted path and ground truth. Note, at inference time, the original DTW problem (i.e., $\gamma = 0$) is solved to generate predicted alignments, leaving a disconnect between the training loss and inference task.

Figure 2: Learning with path information. **Left:** Using Soft-DTW, one can define a loss between the soft, (i.e., $\gamma > 0$) *expected* alignment path against a penalty matrix $\mathbf{\Omega}$. During inference, DTW (i.e., $\gamma = 0$) must be used to produce a predicted alignment. **Right:** Our DecDTW outputs the *optimal* warping path $\phi$ using GDTW at both train *and inference* time, removing the disconnect present in Soft-DTW. DecDTW also allows the regularisation and constraint values to be learnable parameters.

In contrast, DecDTW deviates from the original DTW problem formulation entirely, and uses a continuous-time DTW variant adapted from GDTW (Deriso & Boyd, 2019). The GDTW problem computes a minimum cost, *continuous* alignment path between two *continuous time signals*. We will provide a detailed explanation of GDTW in Section 3. DecDTW allows the *optimal alignment path* $\phi^*$ to be differentiable (w.r.t. layer inputs), as opposed to a soft approximation from Soft-DTW. As a result, the gap between training loss and inference task found in Soft-DTW is not present under DecDTW. In our experiments, we show that using DecDTW to train deep networks to reduce alignment error using ground truth path information greatly improves over Soft-DTW. Figure 2 compares Soft-DTW and DecDTW for learning with path information.

**Deep Declarative Networks**    The declarative framework that allows us to incorporate differentiable optimisation algorithms into a deep network is described in Gould et al. (2021). They present theoretical results and analyses on how to differentiate constrained optimization problems via implicit differentiation. Differentiable convex problems have also been studied recently, including quadratic programs (Amos & Kolter, 2017) and cone programs (Agrawal et al., 2019a;b). This technique has been applied to various application domains including optimisation-based control (Amos et al., 2018), video classification (Fernando & Gould, 2016; 2017), action recognition (Cherian et al., 2017), visual attribute ranking (Santa Cruz et al., 2019), few-shot learning for visual recognition (Lee et al., 2019), and camera pose estimation (Campbell et al., 2020; Chen et al., 2020). To the best of our knowledge, this is the first embedding of an inequality constrained NLP within a declarative layer.

## 3    Continuous Time Formulation for DTW (GDTW)

In this section, we introduce a continuous time version of the DTW problem which is heavily inspired by GDTW (Deriso & Boyd, 2019). The GDTW problem is used to derive the NLP formulation for our DecDTW layer. We will describe how the results in this section can be used to derive a corresponding NLP in Section 4.

### 3.1    Preliminaries

**Signal**    A time-varying signal $\mathbf{x} : [0, 1] \to \mathbb{R}^d$ is a vector-valued function of time. Signals are assumed to be differentiable (at least piecewise smooth) and can be constructed from a time series comprised of observation times $\mathbf{t} \in [0, 1]^N$ where $0 = t_1 < t_2 < \cdots < t_N \leq 1$ and associated observations or *features* given by $\mathbf{X} = \{\mathbf{x}_1, \ldots, \mathbf{x}_N\} \in \mathbb{R}^{N \times d}$, by interpolating between observations (e.g., linear, cubic splines). Without loss of generality, we rescale time for all signals to $[0, 1]$.

**Time warp function**    A time warp function $\phi : [0, 1] \to [0, 1]$ defines correspondences from times in one signal to another, and is similarly assumed to be piecewise smooth. Warp functions typically come with constraints; a common constraint requires that warp functions are non-decreasing, i.e., $\phi'(t) \geq 0$ for all $t$. We interpret $\mathbf{x} \circ \phi$ to be the *time warped* version of $\mathbf{x}$.

**Dynamic time warping**    The GDTW problem can be formulated as follows. Given two signals $\mathbf{x}$ and $\mathbf{y}$, find a warp function $\phi$ such that $\mathbf{y} \circ \phi \approx \mathbf{x}$ in some sense over the time series features. In other words, we wish to bring signals $\mathbf{x}$ and $\mathbf{y}$ together by warping time in $\mathbf{y}$. Figure 1 illustrates

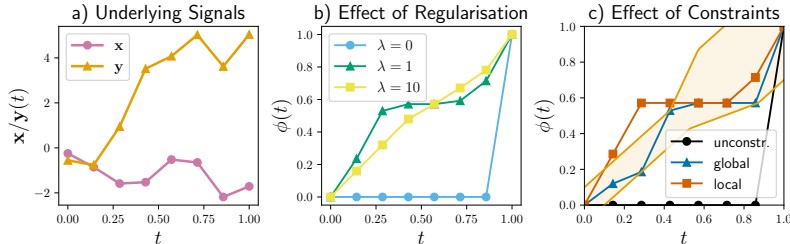

Figure 3: Regularisation (b) and constraints (c) can be applied to prevent pathological warps, e.g., $\lambda = 0$ in (b). Global and local constraints bound warp function values and derivatives, respectively.

how GDTW differs from classic DTW for an example sequence pair. We found in our experiments that GDTW outperformed DTW generally for real-world alignment tasks. We attribute this to the former's ability to align *in between* observations, thus enabling a more accurate alignment.

## 3.2 OPTIMISATION PROBLEM FORMULATION FOR GDTW

GDTW can be specified as a constrained optimisation problem where the decision variable is the time warp function $\phi$ and input parameters are signals $\mathbf{x}$ and $\mathbf{y}$. Concretely, the problem is given by

$$
\begin{aligned}
\text{minimise} \quad & f(\mathbf{x}, \mathbf{y}, \lambda, \phi) \triangleq \mathcal{L}(\mathbf{x}, \mathbf{y}, \phi) + \lambda \mathcal{R}(\phi) \\
\text{subject to} \quad & \underbrace{s^{\min}(t) \leq \phi'(t) \leq s^{\max}(t)}_{\textbf{local constraints}}, \quad \underbrace{b(t)^{\min} \leq \phi(t) \leq b(t)^{\max}}_{\textbf{global constraints}} \quad \forall t.
\end{aligned}
\tag{1}
$$

The objective function $f$ can be decomposed into the *signal loss* $\mathcal{L}$ relating to feature error and a warp regularisation term $\mathcal{R}$, where $\lambda \geq 0$ is a hyperparameter controlling the strength of regularisation. Since $\phi$, $\mathbf{x}$ and $\mathbf{y}$ are functions, our objective function $f$ is a functional, revealing the variational nature of the optimisation problem. We will see in Section 4 that restricting $\phi$ to the space of piecewise linear functions will allow us to easily solve Equation 1 while still allowing for expressive warps.

Furthermore, a number of potentially time-varying constraints are usually imposed on $\phi$. We partition these constraints into *local* and *global* constraints as in Morel et al. (2018). Local constraints bound the time derivatives of the warp function and can be used to enforce the typical nondecreasing warp assumption. Global constraints bound the warp function values directly, with prominent examples being the R-K (Ratanamahatana & Keogh, 2004) and commonly used Sakoe-Chiba (Sakoe & Chiba, 1978) bands. Furthermore, global constraints are used to enforce boundary constraints, i.e., $\phi(t) \in [0, 1]$ for all $t$ and endpoint constraints, i.e., $\phi(0) = 0, \phi(1) = 1$. Not including endpoint constraints allows for *subsequence* GDTW, where a subsequence $\mathbf{y}$ is aligned to only a portion of $\mathbf{x}$. Figure 3 illustrates how constraints can be used to change the resultant warps returned by GDTW.

**Signal Loss** The signal loss functional $\mathcal{L}$ in Equation 1 is defined as

$$
\mathcal{L}(\mathbf{x}, \mathbf{y}, \phi) = \int_0^1 L\big(\mathbf{x}(t) - \mathbf{y}(\phi(t))\big) \mathrm{d}t,
\tag{2}
$$

where $L : \mathbb{R}^d \to \mathbb{R}$ is a twice-differentiable (a.e.) penalty function defined over features. In our experiments we use $L(\cdot) \triangleq \| \cdot \|_2^2$, however other penalty functions (e.g., the 1-norm, Huber loss) can be used. The signal loss (Equation 2) measures the separation of features between $\mathbf{x}$ and time-warped $\mathbf{y}$ and is analogous to the classic DTW discrepancy given by the cost along the optimal warping path.

**Warp Regularisation** The warp regularisation functional $\mathcal{R}$ in Equation 1 is defined as

$$
\mathcal{R}(\phi) = \int_0^1 R\big(\phi'(t) - 1\big) \mathrm{d}t,
\tag{3}
$$

where $R : \mathbb{R} \to \mathbb{R}$ is a penalty function on deviations from the identity warp, i.e., $\phi'(t) = 1$. We use a quadratic penalty $R(u) = u^2$ in this work, consistent with Deriso & Boyd (2019). This regularisation term penalises warp functions with large jumps and a sufficiently high $\lambda$ brings $\phi$ close to the identity. Regularisation is crucial for preventing noisy and/or pathological warps (Zhang et al., 2017; Wang et al., 2016) from being produced from GDTW (and DTW, more generally), and can greatly affect alignment accuracy. We can select $\lambda$ optimally using cross-validation (Deriso & Boyd, 2019).

## 4 SIMPLIFIED NLP FORMULATION FOR GDTW

While Section 3 provides a high-level overview of the GDTW problem, we did not specify a concrete parameterisation for $\phi$, which dictates how the optimisation problem in Equation 3 will be solved. In this section, we provide simplifying assumptions on $\phi$, which allows us to solve Equation 3. We follow the approach in Deriso & Boyd (2019) to reduce the infinite dimensional variational problem in Equation 1 to a finite dimensional NLP. This is achieved by first assuming that $\phi$ is piecewise linear, allowing it to be fully defined by its value at $m$ knot points $0 = t_1 < t_2 < \cdots < t_m = 1$. Knot points can be uniformly spaced or just be the observation times used to parameterise $\mathbf{y}$. Formally, piecewise linearity allows us to represent $\phi$ as $\phi = (\phi_1, \ldots, \phi_m) \in \mathbb{R}^m$ where $\phi_i = \phi(t_i)$. The other crucial assumption involved in reducing Equation 1 to an NLP is replacing the continuous integrals in the signal loss and warp regularisation in Section 3 with numerical approximations given by the trapezoidal rule. These assumptions yield an alternative optimisation problem given by

$$
\begin{aligned}
\text{minimise} \quad & \hat{f}(\mathbf{x}, \mathbf{y}, \lambda, \phi) \triangleq \hat{\mathcal{L}}(\mathbf{x}, \mathbf{y}, \phi) + \lambda \hat{\mathcal{R}}(\phi) \\
\text{subject to} \quad & s_i^{\min} \leq \frac{\phi_{i+1} - \phi_i}{\Delta t_i} \leq s_i^{\max}, \quad b_i^{\min} \leq \phi_i \leq b_i^{\max} \quad \forall i,
\end{aligned}
\tag{4}
$$

where $\Delta t_i = t_{i+1} - t_i$. The new signal loss $\hat{\mathcal{L}}$ is given by

$$
\hat{\mathcal{L}}(\mathbf{x}, \mathbf{y}, \phi) = \sum_{i=1}^{m-1} \frac{L_{i+1} + L_i}{2} \Delta t_i, \quad L_i := L(\mathbf{x}(t_i) - \mathbf{y}(\phi_i)),
\tag{5}
$$

which follows from applying the definition of $\phi$ and the trapezoidal approximation to Equation 2. Note, the non-convexity of objective $\hat{f}$ (even when assuming convex $L$ and $R$) is caused by the $\mathbf{x}(\phi_i)$ terms for arbitrary signal $\mathbf{x}$. The new regularisation term $\hat{\mathcal{R}}$ is given by

$$
\hat{\mathcal{R}}(\phi) = \sum_{i=1}^{m-1} R\left(\frac{\phi_{i+1} - \phi_i}{\Delta t_i} - 1\right) \Delta t_i,
\tag{6}
$$

noting that since $\phi'$ is assumed to be piecewise constant, we can simply use Riemann integration to evaluate the continuous integral in $\mathcal{R}$ exactly. The simplified problem in Equation 4 is actually a continuous non-linear program with decision variable $\phi \in \mathbb{R}^m$ and linear constraints. However, the objective function is non-convex (Deriso & Boyd, 2019) and we will now describe the method used to find good solutions to Equation 4 using approximate dynamic programming.

## 5 DECLARATIVE DTW FORWARD PASS

Our DecDTW layer encodes an implicit function $\phi^\star = \text{DecDTW}(\mathbf{x}, \mathbf{y}, \lambda, \mathbf{s}^{\min}, \mathbf{s}^{\max}, \mathbf{b}^{\min}, \mathbf{b}^{\max})$, which yields the optimal warp given input signals $\mathbf{x}, \mathbf{y}$, regularisation $\lambda$ and constraints $\mathbf{s}^{\min} = \{s_i^{\min}\}_{i=1}^m$, $\mathbf{s}^{\max} = \{s_i^{\max}\}_{i=1}^m$, $\mathbf{b}^{\min} = \{b_i^{\min}\}_{i=1}^m$, $\mathbf{b}^{\max} = \{b_i^{\max}\}_{i=1}^m$. The warp $\phi^\star$ can be used in a downstream loss $J(\phi^\star)$, e.g., the error between predicted warp $\phi^\star$ and a ground truth warp $\phi^{\text{gt}}$. The GDTW discrepancy is recovered by setting $J = \hat{f}$. Compared to Soft-DTW(Cuturi & Blondel, 2017), which produces soft alignments, DecDTW outputs the *optimal*, continuous time warp. The DecDTW forward pass solves the GDTW problem given by Equation 4 given the input parameters. We solve Equation 4 using a dynamic programming (DP) approach as proposed in Deriso & Boyd (2019) instead of calling a general-purpose NLP solver; this is to minimise computation time.

The DP approach uses the fact that $\phi$ lies on a compact subset of $\mathbb{R}^m$ defined by the global bounds, allowing for efficient discretisation. Finding the globally optimal solution to the discretised version of Equation 4 can be done using DP. The resultant solution $\phi^\star$ is an approximation to the solution of the continuous NLP, with the approximation error dependent on the resolution of discretisation. We can reduce the error given a fixed resolution $M$ using multiple iterations of refinement. In each iteration, a tighter discretisation is generated around the previous solution and the new discrete problem solved. A detailed description of the solver can be found in Deriso & Boyd (2019) and the Appendix.

Note, for the purpose of computing gradients in the backward pass, detailed in Section 6, it is important that the approximation $\phi^\star$ is suitably close to the (true) solution of Equation 4. Otherwise, the gradients computed in the backward pass may diverge from the true gradient, causing training to be unstable. The accuracy of $\phi^\star$ is highly dependent on the resolution of discretisation and number of refinement iterations. We discuss how to set DP solver parameters in the Appendix.

## 6 DECLARATIVE DTW BACKWARD PASS

We now derive the analytical solution of the gradients of $\phi^\star$ recovered in the forward pass w.r.t. inputs $\mathbf{z} = \{\mathbf{x}, \mathbf{y}, \lambda, \mathbf{s}^{\min}, \mathbf{s}^{\max}, \mathbf{b}^{\min}, \mathbf{b}^{\max}\} \in \mathbb{R}^n$ using Proposition 4.6 in Gould et al. (2021) (note that gradients w.r.t. signals $\mathbf{x}, \mathbf{y}$ are w.r.t. the underlying observations $\mathbf{X}, \mathbf{Y}$). Unlike existing approaches for differentiable DTW, DecDTW allows the regularisation weight $\lambda$ and constraints to be learnable parameters in a deep network. Let $\tilde{h} = [h_1, \ldots, h_p]$, where each $h_i : \mathbb{R}^n \times \mathbb{R}^m \to \mathbb{R}$ corresponds to an *active* constraint from the full set of inequality constraints described in Equation 4, rearranged in standard form, i.e., $h_i(\cdot) \leq 0$. For instance, an active local constraint relating to time $t_j$ can be expressed as $h_i(\mathbf{z}, \phi) = \phi_{j+1} - \phi_j - \Delta t_j s_j^{\max} \leq 0$. Assuming non-singular $H$ (note, this always holds for norm-squared $L$ and $R$) and $\text{rank}(D_\phi \tilde{h}(\mathbf{z}, \phi)) = p$, the backward pass gradient is given by

$$D\phi(\mathbf{z}) = H^{-1} A^\top (A H^{-1} A^\top)^{-1} (A H^{-1} B - C) - H^{-1} B, \tag{7}$$

where $D\phi(\mathbf{z}) \in \mathbb{R}^{m \times n}$ is the Jacobian of estimated warp $\phi$ w.r.t. inputs $\mathbf{z}$ and

$$\begin{aligned} A &= D_\phi \tilde{h}(\mathbf{z}, \phi) \in \mathbb{R}^{p \times m} & B &= D_{\mathbf{z}\phi}^2 \hat{f}(\mathbf{z}, \phi) \in \mathbb{R}^{m \times n} \\ C &= D_{\mathbf{z}} \tilde{h}(\mathbf{z}, \phi) \in \mathbb{R}^{p \times n} & H &= D_{\phi\phi}^2 \hat{f}(\mathbf{z}, \phi) \in \mathbb{R}^{m \times m}. \end{aligned} \tag{8}$$

Observe that the objective $\hat{f}$ as defined in Equation 4 only depends on a subset of $\mathbf{z}$ corresponding to $\mathbf{x}, \mathbf{y}, \lambda$ and similarly, constraints $\tilde{h}$ only depend on $\mathbf{s}^{\min}, \mathbf{s}^{\max}, \mathbf{b}^{\min}, \mathbf{b}^{\max}$. While Proposition 4.6 in Gould et al. (2021) has additional terms in $B$ and $H$ involving Lagrange multipliers, we note that since $\tilde{h}$ is at most first order w.r.t. $\{\mathbf{z}, \phi\}$, these terms evaluate to be zero and can be ignored. We discuss how to evaluate Equation 7 using vector-Jacobian products efficiently in the Appendix.

## 7 EXPERIMENTS

Our experiments involve two distinct application areas and utilise real-world datasets. For both applications, the goal is to use learning to improve the accuracy of predicted temporal alignments using a training set of labelled ground truth alignments. We will show that DecDTW yields state-of-the-art performance on these tasks. We have implemented DecDTW in PyTorch (Paszke et al., 2019) with open source code available to reproduce all experiments at https://github.com/mingu6/declarativedtw.git.

### 7.1 LEARNING FEATURES FOR AUDIO-TO-SCORE ALIGNMENT

**Problem Formulation** Our first experiment relates to audio-to-score alignment, which is a fundamental problem in music information retrieval (Thickstun et al., 2020; Ewert et al., 2009; Orio et al., 2001; Shalev-Shwartz et al., 2004), with applications ranging from score following to music transcription (Thickstun et al., 2020). The goal of this task is to align an audio recording of a music performance to its corresponding musical score/sheet music. We use the mathematical formulation proposed in Thickstun et al. (2020) for evaluating predicted audio-to-score alignments against a ground truth alignment, which we will now summarise. An alignment is a monotonic function $\phi : [0, S] \to [0, T]$ which maps positions in a score (measured in beats) to a corresponding position in the performance recording (measured in seconds). Two evaluation metrics between a predicted alignment $\phi$ and a ground truth alignment $\phi^{\text{gt}}$ are proposed, namely the temporal average error (TimeErr) and temporal standard deviation (TimeDev), given formally by

$$\text{TimeErr}(\phi, \phi^{\text{gt}}) = \frac{1}{S} \int_0^S |\phi(s) - \phi^{\text{gt}}(s)| \mathrm{d}s, \quad \text{TimeDev}(\phi, \phi^{\text{gt}}) = \sqrt{\frac{1}{S} \int_0^S (\phi(s) - \phi^{\text{gt}}(s))^2 \mathrm{d}s}. \tag{9}$$

Thickstun et al. (2020) assumes alignments are piecewise linear in between changepoints (where the set of notes being played changes) within the score, allowing Equation 9 to be analytically tractable.

**Methodology** Thickstun et al. (2020) provide a method for generating candidate alignments between a performance recording and a musical score as follows: First, synthesize the musical score (using a MIDI file) to an audio recording. Second, extract audio features from the raw audio for both the synthesised score and performance. Third, use DTW to align the two sets of features. Last, use Equation 9 to evaluate the predicted alignment. In these experiments, we learn a feature extractor on top of the base audio features with the goal of improving the alignment accuracy from DTW.

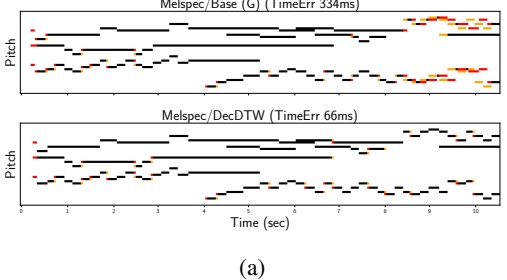
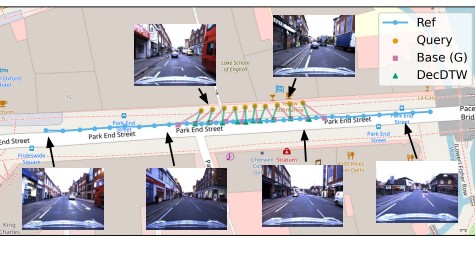

|   (a)   |   (b)   |

Figure 4: **(a)** Example audio-to-score alignment. Red (resp. yellow) identifies notes indicated by audio feature (resp. ground truth) alignments, but not by the ground truth (resp. audio features). DecDTW training yields dramatic improvements for melspec features, even surpassing highly optimised CQT features (38ms TimeErr). **(b)** Example VPR alignment. Query images are aligned to a database sequence using learned image embeddings. Magenta (resp. green) illustrates where the pre-trained (resp. fine-tuned) system estimates the query poses using database geotags. The max (resp. mean) error is reduced from 7.8m to 2.7m (resp. 4.6m to 1.5m) after fine-tuning.

**Experimental Setup**    We use the dataset in Thickstun et al. (2020), which is comprised of 193 piano performances of 64 scores with ground truth alignments taken from a subset of the MAE-STRO (Hawthorne et al., 2019) and Kernscores (Sapp, 2005) datasets. We extract three types of base frequency-domain features from the raw audio; constant-Q transform (CQT), (log-)chromagram (chroma) and (log-)mel spectrogram (melspec). The CQT and chroma features are already highly optimised for pitch compared to melspec features. However, melspec features retain more information from the raw audio compared to CQT and chroma. We evaluated six different methods, described as follows: *1-2)* use base features (no learning) and both DTW (D) and GDTW (G) for alignments. For the remaining methods (3-6), each learns a feature extractor given by a Gated Recurrent Unit (GRU) (Cho et al., 2014), on top of the base features. The learned features are then used to compute the corresponding loss, described as follows: *3)* Soft-DTW (Cuturi & Blondel, 2017) discrepancy loss; *4)* feature matching cost along ground truth alignment path (Along GT); *5)* DILATE Le Guen & Thome (2019), which uses path information through the soft expected alignment under Soft-DTW; and *6)* TimeErr loss training (DecDTW only). For DILATE, we encode the deviation from the ground truth alignment in the path penalty matrix $\Omega$. The full set of hyperparameters, including dataset generation and further information on comparison methods are provided in the Appendix.

Table 1: Summary of results for the audio-to-score alignment experiments

| Reported metrics are TimeErr / TimeDev (ms) | | | | | |
|---|---|---|---|---|---|
| Feature Type | Base (D) | Base (G) | Soft-DTW | Along GT | DILATE | DecDTW |
| CQT | 35 / 90 | 19 / 30 | 49 / 130 | 49 / 130 | 29 / 45 | **17 / 27** |
| Chroma | 50 / 115 | 24 / 39 | 59 / 142 | 59 / 142 | 28 / 41 | **19 / 31** |
| Melspec | 122 / 235 | 56 / 81 | 55 / 153 | 52 / 145 | 26 / 40 | **16 / 27** |

**Results**    A summary of results is presented in Table 1. DTW is used for alignments in Base (D), Soft-DTW, Along GT and DILATE, while GDTW is used for alignments for Base (G) and DecDTW. Training using Soft-DTW fails to improve alignment performance over the validation set, measured by TimeErr, because the ground truth alignment is not used during training[1]. Similarly, minimising the feature matching cost along the ground truth alignment path fails to improve alignment accuracy, since the predicted alignment is not used during training. DILATE, however, significantly improves on Base (D) since it incorporates predicted and ground truth path information during training.

Similarly to DILATE, training using the TimeErr loss with DecDTW significantly improves the alignment accuracy compared to Base (G), effectively utilising both predicted and ground truth path information. DecDTW yields state-of-the-art results overall for all base feature types by a large margin. Note the already the strong performance of Base (G) over both Base (D) and even DILATE

---

[1]The randomly initialised GRU feature extractor also degrades alignment accuracy compared to base features.

(for CQT and chroma); this demonstrates the benefit of using continuous time GDTW alignments. Finally, it is interesting to note that both DILATE and DecDTW are able to utilise the more expressive (versus CQT and chroma) melspec features to surpass the performance of base CQT and chroma. Figure 4a illustrates an example test alignment, with more examples provided in the Appendix.

## 7.2 TRANSFER LEARNING FOR SEQUENCE-BASED VISUAL PLACE RECOGNITION

Our second experiment relates to the Visual Place Recognition (VPR) problem, which is an active area of research in computer vision (Berton et al., 2022; Arandjelovic et al., 2016) and robotics (Garg et al., 2021; Lowry et al., 2015; Cummins & Newman, 2011) and is often an important component of the navigation systems found in mobile robots and autonomous cars. The VPR task involves recognising the approximate geographic location where an image is captured, typically to a tolerance of a few meters (Zaffar et al., 2021; Berton et al., 2022) and is commonly formulated as image retrieval, where a query image is compared to a database of geo-tagged reference images. VPR using image sequences has been shown to improve retrieval performance over single image methods (Garg & Milford, 2021; Xu et al., 2020; Stenborg et al., 2020). In this experiment we first formulate sequence-based VPR as a temporal alignment problem and then fine-tune a deep feature extractor for this alignment task.

**Problem formulation** Let $\{I_i^r\}_{i=1}^N$, $\mathbf{t}^r \in \mathbb{R}^N$ and $\{\mathbf{x}_i^r\}_{i=1}^N$ be a reference image sequence, associated (sorted) timestamps and geotags, respectively. Similarly, let $\{I_i^q\}_{i=1}^M$, $\mathbf{t}^q \in \mathbb{R}^M$ and $\mathbf{X}^q = \{\mathbf{x}_i^q\}_{i=1}^N$ be the equivalent for a query sequence, noting that query geotags are used only for training and evaluation. Furthermore, when using deep networks for VPR, we have a feature extractor $f_\theta$ with weights $\theta$ that is used to extract embeddings $\mathbf{Z}^r = \{f_\theta(I_i^r)\}_{i=1}^N \in \mathbb{R}^{N \times d}$ and $\mathbf{Z}^q = \{f_\theta(I_i^q)\}_{i=1}^M \in \mathbb{R}^{M \times d}$ from reference and query images, respectively. Finally, let $\mathbf{x}^r$, $\mathbf{x}^q$, $\mathbf{z}^r$, $\mathbf{z}^q$ be continuous time signals built from geotags and image embeddings using associated timestamps.

We can formulate sequence-based VPR as a temporal alignment problem, where the objective is to minimise the discrepancy between the predicted alignment derived from the query and reference image embeddings and the optimal alignment derived from the geotags. Concretely, the goal of our learning problem is to minimise w.r.t. network weights $\theta$, a measure of localisation error $G(\mathbf{x}^r(\phi), \mathbf{X}^q)$, where $\phi = \text{DTW}(\mathbf{z}^r, \mathbf{z}^q; \theta)$ defined as $\phi_i = \phi(t_i^q)$ is the estimated alignment which maps query images to positions in the reference sequence and $\mathbf{x}^r(\phi) = \{\mathbf{x}^r(\phi_i)\}_{i=1}^M$. We select $G$ as the (squared) maximum error over the queries, given by $G(\mathbf{X}^1, \mathbf{X}^2) = \max\{\|\mathbf{x}_i^1 - \mathbf{x}_i^2\|_2^2 : i = 1, \ldots, M\}$.

**Experimental setup** We source images and geotags from the Oxford RobotCar dataset (Maddern et al., 2017), commonly used as a benchmark to evaluate VPR methods. This dataset is comprised of autonomous vehicle traverses of the same route captured over two years across varying times of day and environmental conditions. We use a subset of $\sim$1M images captured across 50+ traverses provided by Thoma et al. (2020) with geotags taken from RTK GPS (Maddern et al., 2020) where available, accurate to 15cm. These images are split into training, validation and test sets, which are used to train a base network for single image retrieval as well as our sequence-based methods. The train and validation sets are captured in the same geographic areas on distinct dates whereas the test set is captured in an area geographically disjoint from validation and training.

Paired sequences are generated using the GPS ground truth; reference sequences have 25 images spaced $\sim$5m apart and query sequences have 10 images sampled at $\sim$1Hz. This setup is close to a deployment scenario where geotags are available for the reference images and query images exhibit large changes in velocity compared to the reference. Subsequence GDTW is used to align queries to the reference sequence. In total, we use $\sim$22k, $\sim$4k, $\sim$1.7k sequence pairs for training, validation and testing, respectively. Our test set contains a diverse set of sequences over a large geographic area, and exhibits lighting, seasonal and structural change between reference and query. We provide our paired sequence dataset in the supplementary material. Finally, for feature extraction, we use a VGG backbone with NetVLAD aggregation layer (Arandjelovic et al., 2016) (yields 32k-dim embeddings) and train the single image network using triplet margin loss. We fine-tune the conv5 and NetVLAD layers for sequence training. See the Appendix for a detailed description of the training setup.

**Results** We evaluate all methods by measuring the proportion of test sequences where the maximum error is below predefined distance thresholds; these metrics are commonly used for single image methods (Garg et al., 2021; Berton et al., 2022). Results are split into three different environmental conditions including overcast, sunny and snow. We present results for two methods using the base single image network (1-2) and four methods which perform fine-tuning of the base network for

sequence VPR (3-6): *1-2)* Base features (no sequence fine-tuning) with DTW (D) and GDTW (G) alignment, *3)* OTAM Cao et al. (2020) (Soft-DTW w/subsequence alignment) discrepancy, *4)* Feature cost along the ATE minimising alignment (Along GT), *5)* DILATE loss and *6)* max error loss (DecDTW). Similar to the audio experiments, DTW was used to produce alignments for methods (3-5) and GDTW was used for (6). For DILATE, we again modified the path penalty matrix $\Omega$ to encode the deviation from the ground truth path. Table 2 summarises the results.

Table 2: VPR experiment results (maximum error): Test Accuracy @2/3/5/10m

| Method | Overcast | Sunny | Snow |
|---|---|---|---|
| Base (D) | 0.7/12.3/44.3/73.8 | 0.5/8.7/40.2/62.2 | 1.6/11.1/44.3/62.3 |
| Base (G) | 10.2/31.7/65.6/**90.3** | 9.0/33.2/75.1/**99.3** | 6.2/25.7/73.2/**100.0** |
| OTAM/Along GT | 0.7/12.3/44.3/73.8 | 0.5/8.7/40.2/62.2 | 1.6/11.1/44.3/62.3 |
| DILATE | 0.7/14.5/54.5/80.0 | 1.5/11.9/40.9/69.7 | 1.3/12.2/44.6/68.7 |
| DecDTW | **25.4/45.0/68.3**/90.1 | **22.8/50.4/84.3**/98.3 | **13.5/44.8/88.0/100.0** |

We found that fine-tuning with the OTAM and Along GT losses is not able to reduce localisation error over the validation set compared to using base features. This is because the OTAM loss does not use GPS ground truth and the Along GT loss does not use the predicted alignment. DILATE, which utilises path information, improves over Base (D), especially for the 5-10m error tolerances. However, training with DecDTW to directly reduce the maximum GPS error between predicted and ground truth yields *significant* improvements compared to Base (G) across tighter ($\leq$ 5m) error tolerances. Similar to the audio experiments, we see the benefit of using GDTW alignments over DTW when comparing Base (G) to Base (D) and DILATE. However, DecDTW improves over Base (G) substantially more compared to DILATE over Base (D) when compared to the audio experiments. We provide more results and analysis for the VPR task in the Appendix. Figure 4b illustrates an example of a test sequence before and after fine-tuning, with more examples provided in the Appendix.

## 7.3 SCALABILITY

Figure 5 illustrates how inference at test time scales with time series length $n$ for GDTW (required for DecDTW) compared to DTW. Timings are evaluated on an Nvidia GeForce GTX 1080Ti 11Gb. GDTW is between 15-50 times slower than DTW, with the gap increasing with length $n$. The additional computation is due in large part to the additional discretisation required to solve for the finer-grained continuous time problem defined in Section 4. DTW solves a DP with $O(mn)$ nodes and $O(mn)$ edges, whereas GDTW solves a DP with $O(mM)$ edges and $O(mM^2)$ edges (we set $M = n$). However, by removing iterations of warp refinement, we can reduce this gap to a 4-12x slowdown with an mean accuracy loss of only 0.5%. Further discussion and a comparison of train-time scalability is provided in the Appendix.

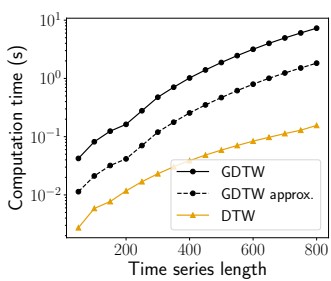

Figure 5: Test time scalability

## 8 CONCLUSION AND FUTURE WORKS

We present a novel formulation for a differentiable DTW layer that can be embedded into a deep network, based on continuous time warping and implicit deep layers. Unlike existing approaches, DecDTW yields the *optimal* alignment path as a differentiable function of layer inputs. We demonstrate the effectiveness of DecDTW in utilising path information in challenging, real-world alignment tasks compared to existing methods. One promising direction for future work is in exploring more compact, alternative parameterisations of the warp function to improve scalability, inspired by Zhou & De la Torre (2012). Another direction would be to integrate more sophisticated DTW variants such as jumpDTW (Fremerey et al., 2010), which allows for repeats and discontinuities in the warp, into the GDTW (and DecDTW) formulation. Finally, we presented methodology for allowing regularisation and constraints to be learnable parameters in a deep network but did not explore this in our experiments. Future work will also explore this capability in more detail.

ACKNOWLEDGMENTS

S. Garg and M. Milford are with the QUT Centre for Robotics, School of Electrical Engineering and Robotics at the Queensland University of Technology. M. Xu and S. Gould are with the School of Computing, College of Engineering, Computing and Cybernetics at the Australian National University. The majority of the work was completed while M. Xu was at the QUT Centre for Robotics. M. Xu was supported by an Australian Government Research Training Program (RTP) Scholarship as well as by the QUT Centre for Robotics. M. Milford is supported by funding from ARC Laureate Fellowship (FL210100156) funded by the Australian Government. S. Gould is the recipient of an ARC Future Fellowship (FT200100421) funded by the Australian Government. Finally, we thank the reviewers and the area chair for their insightful and constructive comments during the discussion period.

REPRODUCIBILITY STATEMENT

To facilitate reproducibility, we provide a reference implementation for our method, including code to reproduce all results across both of the experiments presented in the paper. Our code provides the full training and evaluation pipeline for all methods. In addition, we provide detailed instructions for setting up the datasets for training and evaluation, again for both experiments, as per the descriptions provided in the Appendix. Furthermore, we make publicly available the model checkpoint for the pre-trained single-image baseline network used for sequence fine-tuning in the VPR experiments.

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

# A    APPENDIX

We provide additional implementation details for DecDTW, as well as additional qualitative examples for both experiments in this section.

## A.1    DYNAMIC PROGRAMMING SOLVER FOR GDTW

The mechanics of the solver are described as follows. For each $i$ we can discretise $\phi_i$ into $M$ values $\{\phi_{i,j}\}_{j=1}^M$ uniformly between global bounds $b_i^{\min}, b_i^{\max}$. This forms a graph where each of the $mM$ values correspond to nodes and furthermore, temporally adjacent nodes, i.e., $\phi_{i-1,j}, \phi_{i,k}$ for all $j, k$ are connected with edges, for a total of $(m-1) \times M^2$ edges. Node costs correspond to signal loss values in Equation 5 and edge costs correspond to warp regularisation values in Equation 6. Edges which violate the local constraints $s_i^{\min}, s_i^{\max}$ are given a cost of $\infty$. The global minima of this new discrete optimisation problem corresponds to the minimum cost path through the graph and is solved using dynamic programming in $O(mM^2)$ time complexity. Iterative refinement of the discretisation and solution is performed as described in Deriso & Boyd (2019). We use three iterations of discretisation with discretisation factor $\eta = 0.125$ in all of our experiments. Furthermore, we set $M = \max(50, n)$, where $n$ is the length of time series $\mathbf{X}$. Figure 6 illustrates the mechanics of the DP solver for a subsequence alignment problem and shows that the solution after refinement is comparable to calling an SLSQP solver (Virtanen et al., 2020) directly.

In general, one can set $\eta$ and $M$ using hyperparameter optimisation over a validation dataset. Assume we have a validation dataset where each observation is comprised of a time series pair. For any given $\eta$ and $M$ and number of refinement iterations $r$, we can use GDTW to compute the optimal alignment $\phi^\star$ using the DP solver. Furthermore, we can use $\phi^\star$ to initialise an NLP solver such as in Virtanen et al. (2020) to find a refined solution $\phi_{NLP}^\star$, which will be closer to the true solution of Equation 4. The goal of the hyperparameter optimisation is to find the least computationally expensive solver configuration (measured by computation time) subject to constraints on approximation error (measured w.r.t. objective function $\hat{f}$ or warp $\phi$).

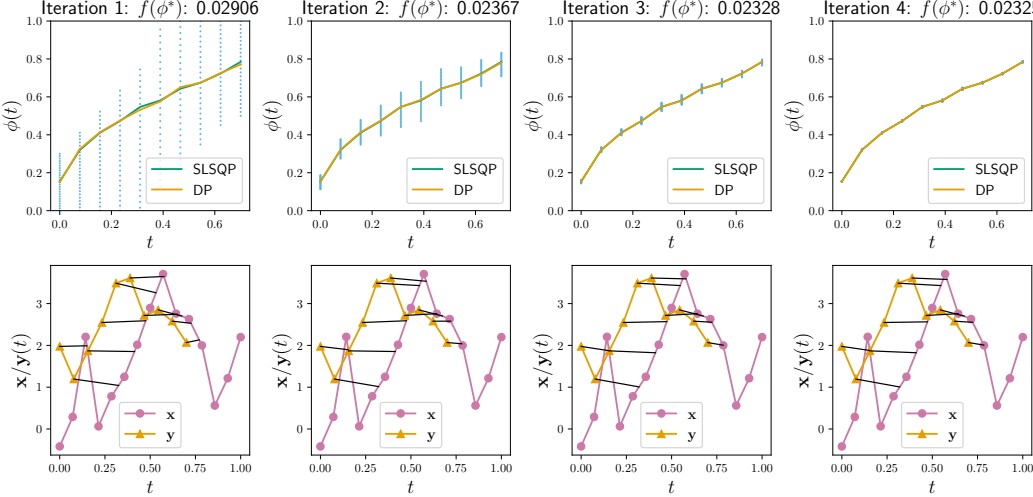

Figure 6: Illustration of the DP solver mechanics on a subsequence alignment problem, where $\phi$ maps times in the shorter sequence $\mathbf{y}$ to longer sequence $\mathbf{x}$. A uniform discretisation is generated between global constraints in the first iteration. Subsequent discretisations are made between tighter bounds around the previous solution. The objective function value decreases at each iteration until convergence after 3 iterations. The objective value $\hat{f}$ from the SLSQP solver (Virtanen et al., 2020) is 0.02325, close to the DP solver at the final iteration and furthermore, we can see qualitatively that the solved warps are indistinguishable between the DP and SLSQP approach.

## A.2 Efficient Computation of Backward Pass

We do not construct the full Jacobian $D\phi(\mathbf{z})$ from Equation 7 explicitly, instead directly computing the vector-Jacobian products required for automatic differentiation libraries. Let $J(\phi)$ be a downstream loss function defined over the estimated alignment (e.g., MSE to a ground truth alignment $\phi^{\text{gt}}$). Our goal for the end-to-end model is to compute $DJ(\mathbf{z}) = v^\top D\phi(\mathbf{z})$, where $v^\top = J(\phi) \in \mathbb{R}^{1 \times m}$. To do this, we evaluate $v^\top D\phi(\mathbf{z})$ from left to right, pre-computing and caching $H^{-1} A^\top$ beforehand. In addition, note that $H$ is tridiagonal from the definition of $\hat{f}$, where off-diagonal elements correspond only to the warp regularisation component $\hat{\mathcal{R}}$. This allows us to solve $v^\top H^{-1}$ in $O(m)$ time. Finally, observe that $B$ and $C$ are in a sense complementary to each other. Specifically, columns of $B$ relating to $\mathbf{s}^{\min}, \mathbf{s}^{\max}, \mathbf{b}^{\min}, \mathbf{b}^{\max}$ are zero since the objective function $\hat{f}$ only depends on $\mathbf{x}, \mathbf{y}, \lambda$. Conversely, columns of $C$ relating to $\mathbf{x}, \mathbf{y}, \lambda$ are zero since the constraint functions only depend on $\phi$ and $\mathbf{s}^{\min}, \mathbf{s}^{\max}, \mathbf{b}^{\min}, \mathbf{b}^{\max}$. This allows efficient evaluation of Equation 7 w.r.t the two blocks $\{\mathbf{x}, \mathbf{y}, \lambda\}$ and $\{\mathbf{s}^{\min}, \mathbf{s}^{\max}, \mathbf{b}^{\min}, \mathbf{b}^{\max}\}$ separately by setting $C$ (resp. $B$) to zero. Our full, efficient backward pass implementation as decribed is provided in our code at https://github.com/mingu6/declarativedtw.git.

## A.3 Train Time Scalability Analysis

We present results for train time scalability in this section using the same experimental setup as the test time scalability analyisis presented in Section 5. Unlike at test time, we aren't able to trade-off warp estimation accuracy for computation time during training. This is because the backward pass computation given by Equation 7, assumes the estimated warp $\phi^\star$ satisfies the first-order optimality conditions for the NLP defined in Equation 4. To ensure this assumption holds, we need to ensure our solver discretises $\phi$ finely enough such that $\phi^\star$ is sufficiently close to an optimal point (up to a suitable numerical tolerance). In practice, this can be achieved with multiple rounds of iterative refinement (we use three in our experiments, as described in Section 6) and a sufficiently large discretistion level $M$ (again, we set $M = n$ in this analysis).

Figure 7 illustrates the results of our train time scalability analysis. Note, the computation times presented include the *total* time taken for a training iteration, including the forward and backward pass. We observe that a training iteration of DecDTW is 4-20x (resp. 15-50x) slower than Soft-DTW (resp. DTWNet), due to the iterative warp refinement required for fine-grained warp estimation in the DecDTW forward pass. However, we observe that DI-LATE is in fact slower for training compared to DecDTW; this is attributed to the relevant efficiency of our backward pass being explicitly computed rather than computed through unrolling the DP recursion.

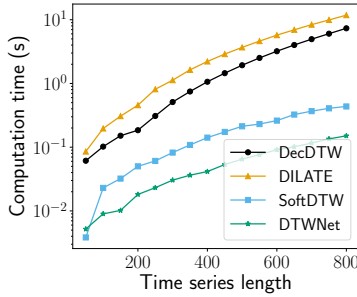

Figure 7: Train time scalability

## A.4 Experimental Setup for Audio-to-Score Alignment

**Dataset Description** Ground truth alignments were generated between the 193 piano performances to their associated scores using the methodology proposed in Thickstun et al. (2020). We used the authors' default hyperparameters for ground truth generation. These 193 performances are divided into train (94), validation (49) and test (50) splits. Furthermore, scores are not shared across splits. Audio is sampled at 22.1kHz with a hop size of 1024 for generating a uniformly spaced time series of base features for each performance and synthesised score. Furthermore, each performance is split into uniform, non-overlapping slices of length 256, which corresponds to ∼12 seconds of audio per slice. There were a total of 995, 565 and 541 slices, across the train, validation and test splits, respectively. The hyperparameters used to generate constant-Q transform (48-dim) and chromagram (12-dim) features are identical to the ones used in Thickstun et al. (2020). For the mel spectrogram features, we used 128 mel bands, yielding a 128-dim feature. We provide the full dataset generation procedure in our code.

**Learning Task** For learned methods (3-6), we apply a feature extractor comprised of a single layer bi-directional GRU Cho et al. (2014) (with random initialisation) on top of base audio features. This feature extractor operates as a sequence-to-sequence model. Specifically, the GRU layer takes as input

a length $N$ sequence of base audio features of shape $\mathbb{R}^{N \times d}$, and outputs learned features of shape $\mathbb{R}^{N \times d_1}$, where the output dimension of the GRU layer is set as $d_1 = 128$ in our experiments. The learned features are then L2-normalised before being used to compute both a downstream alignment and training loss for each learned comparison method. Alignments in Table 1 are generated using DTW for Base (D), Soft-DTW, Along GT, DILATE and using GDTW for Base (G) and DecDTW.

**Comparison Methods**    The training losses for (3-6) are described as follows: The Soft-DTW (Cuturi & Blondel, 2017) method (3) computes the Soft-DTW discrepancy between two feature sets and utilises no path information. Along GT (4) computes the discrepancy between features along the path given by ground truth alignments. DILATE (Le Guen & Thome, 2019) (5) is a hybrid loss which consists of a *shape loss* (equivalent to the Soft-DTW discrepancy) and a *temporal loss*, which penalises deviations from the predicted and ground truth paths. The relative weighting between both losses is given by a hyperparameter $0 \leq \alpha \leq 1$. Ground truth path information in DILATE is given by a penalty matrix $\Omega$. Elements within each row (corresponding to a single point in time in the score) of $\Omega$ (where ground truth alignments are defined) is given a penalty term based on the absolute alignment error. DecDTW (6) is able to train on the $\mathrm{TimeErr}$ loss directly. Full implementation for DecDTW and all comparison methods for this experiment are provided in our code.

**Training Hyperparameters**    We use the Adam (Kingma & Ba, 2015) optimiser with a learning rate of 0.0001 and a batch size of 5 for all methods. We trained for 300 epochs for DILATE and 20 epochs for all remaining methods and selected the model which yielded minimum $\mathrm{TimeErr}$ over the validation set. For DecDTW and Base (G), we set GDTW the regularisation hyperparameter $\lambda = 0.1, 0.7, 0.9$ for CQT, chroma and melspec features, respectively. These regularisation weight values were selected using a grid search over the average $\mathrm{TimeErr}$ across the validation set for the base features. For the Soft-DTW (Cuturi & Blondel, 2017) comparison method we used $\gamma = 1$, noting that results do not meaningfully change for different $\gamma$. For DILATE, $\alpha = 0$ and $\gamma = 0.1$ yielded the best results. We found that a low learning rate and high number of epochs were required for stable training under DILATE, in contrast to DecDTW.

## A.5    EXPERIMENTAL SETUP FOR VISUAL PLACE RECOGNITION

**Construction of Paired Sequences Dataset**    The single-image dataset used to train the baseline VPR retrieval network is used to bootstrap the paired sequences dataset used for sequence-based fine-tuning. Within each train/val/test split, GPS data is used to produce sequence pairs. We designated a fixed set of reference traverses (6 for training, 1 each for validation and test), with each traverse having an associated full data collection run with a given date/time identifier (e.g., 2015-08-17-13-30-19). Each reference traverse is split into geographically overlapping but distinct reference sequences of length 25, with images spaced 5 meters apart within each reference sequence. Adjacent reference sequences within the same traverse have 20 meters overlap between them for training and 12 meters overlap for validation and test splits. For each reference sequence, five query sequences are sampled. Query sequences are randomly selected from the pool of available remaining (i.e. non-reference) traverses within each split. The starting location of each query sequence is chosen randomly such that the query sequence is geographically contained by the reference sequences.

Our resultant dataset, comprised of ~22k, ~4k, ~1.3k sequence pairs for training, validation and testing, respectively, full covers the geographic region and a large set of conditions encompassed by the RobotCar dataset. A large variety of sequence pairs spanning different locations and condition pairs between query and reference enables features learned by sequence fine-tuning to generalise between training and the unseen (both in geographic location and date) test set. We also ensure that the large number of and uniformly spaced test sequences allow for a comprehensive and challenging evaluation. While we can arbitrarily increase the size of this dataset, we find the variety present in our bootstrapped dataset is enough to yield significant gains for sequence fine-tuning. Full dataset lists for the paired sequence datasets are provided in the supplementary material.

**Comparison Methods**    Similar to the audio experiments, all learned sequence fine-tuning methods apply a loss on top of sequences of extracted image embeddings. The OTAM (Cao et al., 2020) loss is the Soft-DTW discrepancy loss with a minor modification to allow for subsequence alignment. Along GT computes the discrepancy between features along the path given by ground truth alignments. For DILATE (Le Guen & Thome, 2019), similar to the audio experiments, we needed to parameterise

the penalty matrix $\Omega$. The value for element $i, j$ is given by the (squared) GPS error (in meters) between the geotages of query image $i$ and reference image $j$. Finally, DecDTW is able to use the *task loss*, i.e. the *maximum* (or mean) GPS error along the *optimal* alignment to improve localisation performance. DILATE is not able to learn directly on the task loss due to the path being encoded as a *soft* alignment path matrix.

**Hyperparameters for Single Image VPR**  We train single image NetVLAD representations with VGG-16 backbone pretrained on ImageNet. NetVLAD layer uses 64 clusters, leading to a 32,768 dimensional time series embedding. Images are resized to $240 \times 180$ (width $\times$ height), consistent with Thoma et al. (2020). Triplet loss margin is set to 0.1 and 10 negatives per anchor-positive pair are used. The model is trained using the Adam optimizer with a learning rate of 1e-5 and a batch size of 4. Our training setup is based on the recently released visual geolocalization benchmark (Berton et al., 2022) and its associated codebase[2]. We provide training and dataset setup code in the supplementary materials.

**Hyperparameters for Sequence-based VPR**  For the sequence fine-tuning experiments, we use a learning rate of 0.0001, batch size of 8 for a maximum of 10 epochs across all methods. In contrast to the audio experiments, query and reference images are both fed through a shared feature extractor comprised of a VGG-16 backbone, followed by a NetVLAD aggregation layer. We checkpoint models during training based on the average maximum GPS error over the full validation set five times per epoch. We selected a fixed regularisation weight of $\lambda = 0.1$ for DecDTW using the validation set and embeddings from the single-image model. For OTAM Cao et al. (2020), we selected $\gamma = 1$, similar to the audio experiments, noting results were insensitive to this parameter. For DILATE Le Guen & Thome (2019), we used $\gamma = 0.1$ and similar to the audio experiments, we set $\alpha = 0$ and $\gamma = 1$.

### A.6  Additional Results for Audio-to-Score Alignment Experiments

In this section, we investigate if features learned using each comparison method presented in Section 7.1 generalises across both DTW and GDTW alignments at test time. In Table 3, we present results for each comparison method using only DTW alignments at test time. Notably, the main difference to Table 1 is the DecDTW column; the features output by the GRU(s) trained using TimeErr loss are placed into a DTW alignment layer (as opposed to GDTW) during testing.

We can see in Table 3 that features learned with DecDTW produce poor alignment accuracy using DTW for both CQT and chroma features when compared to both base features (no learning) and DILATE. In addition, test alignment performance using DTW is significantly poorer compared to using GDTW (see last column in Table 4).

Table 3: Results for the audio-to-score alignment experiments (DTW alignment)

| Reported metrics are TimeErr / TimeDev (ms) | | | | | |
|---|---|---|---|---|---|
| Feature Type | Base (D) | Soft-DTW | Along GT | DILATE | DecDTW |
| CQT | 35/ 90 | 49 / 130 | 49 / 130 | **29 / 45** | 100 / 210 |
| Chroma | 50 / 115 | 59 / 142 | 59 / 142 | **28 /41** | 45 / 117 |
| Melspec | 122 / 235 | 55 / 153 | 52 / 145 | **26 / 40** | 31 / 76 |

We also present results in Table 4, which evaluate features learned under each comparison method using only GDTW alignments at test time. Features learned with methods based on DTW (i.e. Soft-DTW, Along GT, DILATE) perform worse than base features for GDTW alignment. This holds for especially true for DILATE, which yielded accuracy gains when evaluating with DTW alignments compared to base features but significantly degraded accuracy for GDTW alignment. DecDTW as expected, yielded gains compared to base features when evaluated using GDTW alignments.

Overall, we conclude that features trained with an underlying alignment method (either DTW or GDTW) tends to specialise to the particular alignment method used in training. Note that the continuous time formulation of GDTW, which allows for interpolation of alignments, and the availability of regularisation $\lambda$, cause GDTW alignments to outperform DTW overall at test time across most combinations of methods and features.

---

[2]https://github.com/gmberton/deep-visual-geo-localization-benchmark

Table 4: Results for the audio-to-score alignment experiments (GDTW alignment)

| | Reported metrics are TimeErr / TimeDev (ms) | | | | |
|---|---|---|---|---|---|
| Feature Type | Base (G) | Soft-DTW | Along GT | DILATE | DecDTW |
| CQT | 19 / 30 | 22 / 33 | 22 / 33 | 43 / 69 | **17 / 27** |
| Chroma | 24 / 39 | 34 / 51 | 34 / 51 | 70 / 112 | **19 / 31** |
| Melspec | 56 / 81 | 155 / 219 | 147 / 211 | 43 / 68 | **16 / 27** |

## A.7 ADDITIONAL RESULTS FOR VISUAL PLACE RECOGNITION EXPERIMENTS

In this section, we provide additional results to the ones presented in Table 2 in Table 5. First, we add results for the *independent retrieval* task. The independent retrieval task involves finding the most visually similar reference image for each query using nearest neighbours, independently, using image embeddings (no DTW optimisation). Interestingly, fine-tuning a network trained using contrastive learning on single images (this loss is designed to maximise single image retrieval performance) using a sequence-based loss tends to overall improve independent retrieval performance at finer error thresholds ($\leq$ 5m). However, performance is reduced at the looser tolerances ($>$ 5m). This holds uniformly and significantly for DecDTW and DILATE across all test conditions. We hypothesise that the embeddings of networks fine-tuned on the alignment task are trained to better differentiate between more nuanced visual details between nearby reference images.

Table 5: VPR experiment results (maximum error): Test Accuracy @2/3/5/10m

| Method | Overcast | Sunny | Snow |
|---|---|---|---|
| Base (DTW) | 0.7/12.3/44.3/73.8 | 0.5/8.7/40.2/62.2 | 1.6/11.1/44.3/62.3 |
| Base (GDTW) | 10.2/31.7/65.6/**90.3** | 9.0/33.2/75.1/**99.3** | 6.2/25.7/73.2/**100.0** |
| OTAM/Along GT | 0.7/12.3/44.3/73.8 | 0.5/8.7/40.2/62.2 | 1.6/11.1/44.3/62.3 |
| DILATE | 0.7/14.5/54.5/80.0 | 1.5/11.9/40.9/69.7 | 1.3/12.2/44.6/68.7 |
| DecDTW | **25.4/45.0/68.3**/90.1 | **22.8/50.4/84.3**/98.3 | **13.5/44.8/88.0/100.0** |
| Base (Indep.) | 0.0/11.6/47.9/77.0 | 0.0/7.5/40.4/81.1 | 1.6/10.0/50.1/87.1 |
| DILATE (Indep.) | 0.2/11.1/58.4/84.7 | 0.2/17.2/61.7/93.0 | 1.6/13.7/66.3/96.2 |
| DecDTW (Indep.) | 0.2/14.5/50.1/65.6 | 0.2/9.9/35.8/56.7 | 0.9/11.1/57.7/85.8 |

We additionally present results for the *mean* pose error across the sequence (as opposed to the maximum in Table 5) in Table 6. This measure of positioning error is commonly referred to as the absolute trajectory error (ATE). The conclusions are equivalent to the maximum error case; fine-tuning on the GPS error directly using our DecDTW layer yields considerable performance improvements at sub 5m tolerances over the comparison methods.

Table 6: Additional VPR experiment results (mean error): Test Accuracy @2/3/5/10m

| Method | Overcast | Sunny | Snow |
|---|---|---|---|
| Base (DTW) | 16.9/48.9/70.0/94.7 | 16.2/42.9/62.0/91.8 | 16.2/42.9/62.0/91.8 |
| Base (GDTW) | 44.8/70.2/90.1/**98.1** | 47.0/85.2/**99.5/99.8** | 43.5/79.2/**100.0/100.0** |
| OTAM/Along GT | 16.9/48.9/70.0/94.7 | 46.2/42.9/62.0/91.8 | 16.2/42.9/62.0/91.8 |
| DILATE | 22.5/57.9/77.0/96.1 | 19.1/43.1/69.5/93.2 | 15.1/40.4/67.0/96.5 |
| DecDTW | **52.1/80.2/91.8**/97.8 | **65.9/90.8**/98.8/**99.8** | **56.5/93.4/100.0/100.0** |
| Base (Indep.) | 18.6/55.7/80.1/91.5 | 16.9/50.8/85.4/96.8 | 17.0/58.5/90.9/98.4 |
| DILATE (Indep.) | 26.9/69.7/90.1/93.7 | 32.9/69.0/94.2/**99.8** | 22.8/68.5/97.3/99.1 |
| DecDTW (Indep.) | 22.5/55.9/72.4/85.7 | 26.2/59.8/72.6/85.7 | 19.3/61.4/86.0/92.7 |

## A.8 ADDITIONAL QUALITATIVE EXAMPLES FOR AUDIO-TO-SCORE ALIGNMENT EXPERIMENTS

In Figure 8 we present additional visualisations of example alignments for the audio to score alignment task. All features types (CQT, chroma, melspec) and before/after learning are presented in the figures.

Example alignments are randomly sampled from the test set. The visualisations are a way proposed by Thickstun et al. (2020) to visually compare two alignments (e.g., an estimated and ground truth alignment) by plotting *performance aligned scores*, where a performance aligned score is generated by mapping the score through a warping function.

We generate a performance aligned score using both the ground truth alignment and the estimated alignments using audio features. Red identifies notes indicated by audio feature alignments, but not by the ground truth, and yellow identifies notes indicated by the ground truth, but not by the audio features. Note, even if a predicted alignment is close to the ground truth (in a $\mathrm{TimeErr}$ sense), mistakes made by the performer may yield significant looking errors (orange and red) in the visualisations.

### A.9 ADDITIONAL QUALITATIVE EXAMPLES FOR VISUAL PLACE RECOGNITION EXPERIMENTS

We provide additional qualitative examples for the VPR experiments in Figure 9. These figures illustrate the effect of DecDTW fine-tuning on resultant positioning accuracy for a set of test sequence pairs. In addition, Figure 9 provides example test images to illustrate the amount of appearance change observed between reference and query images.

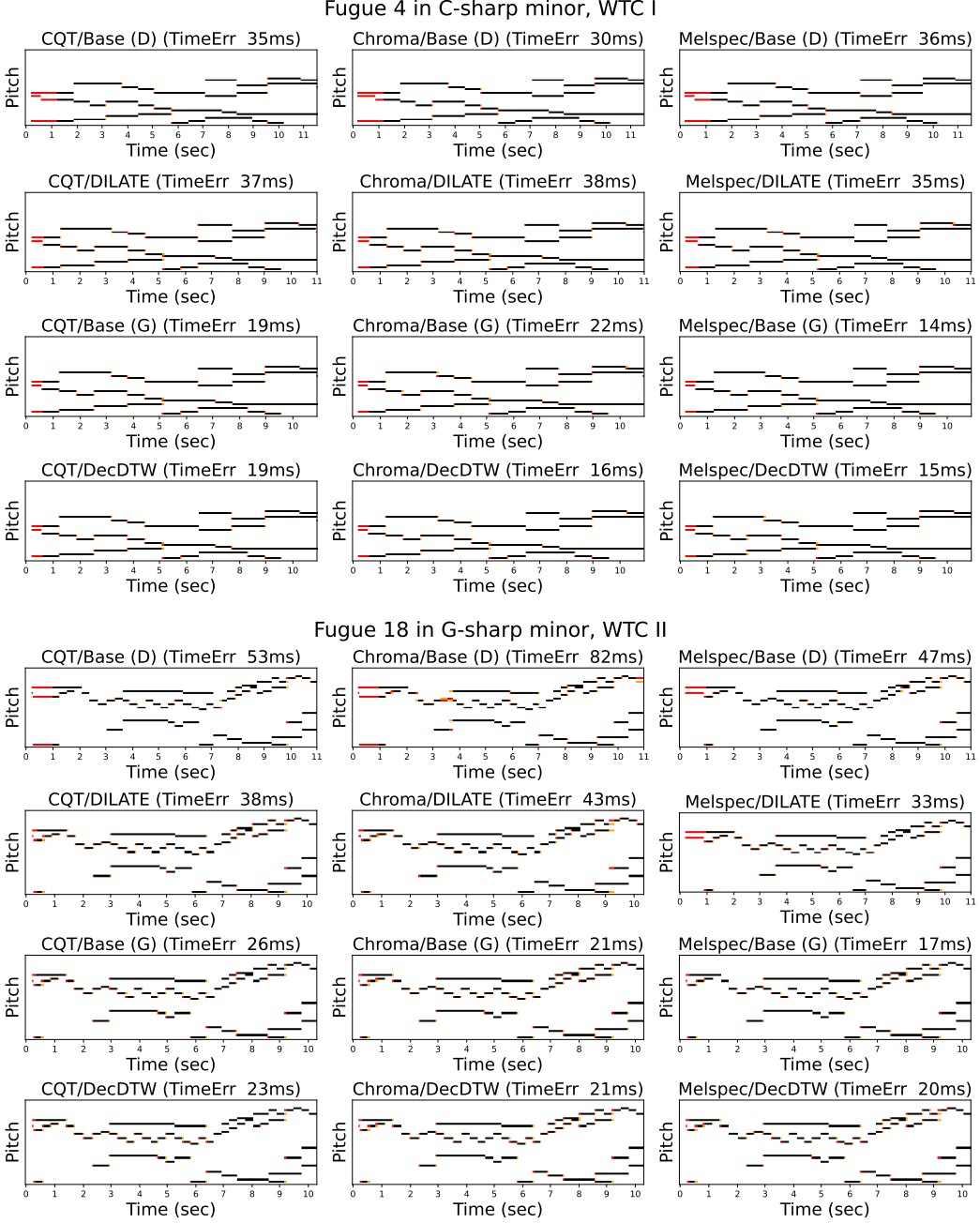

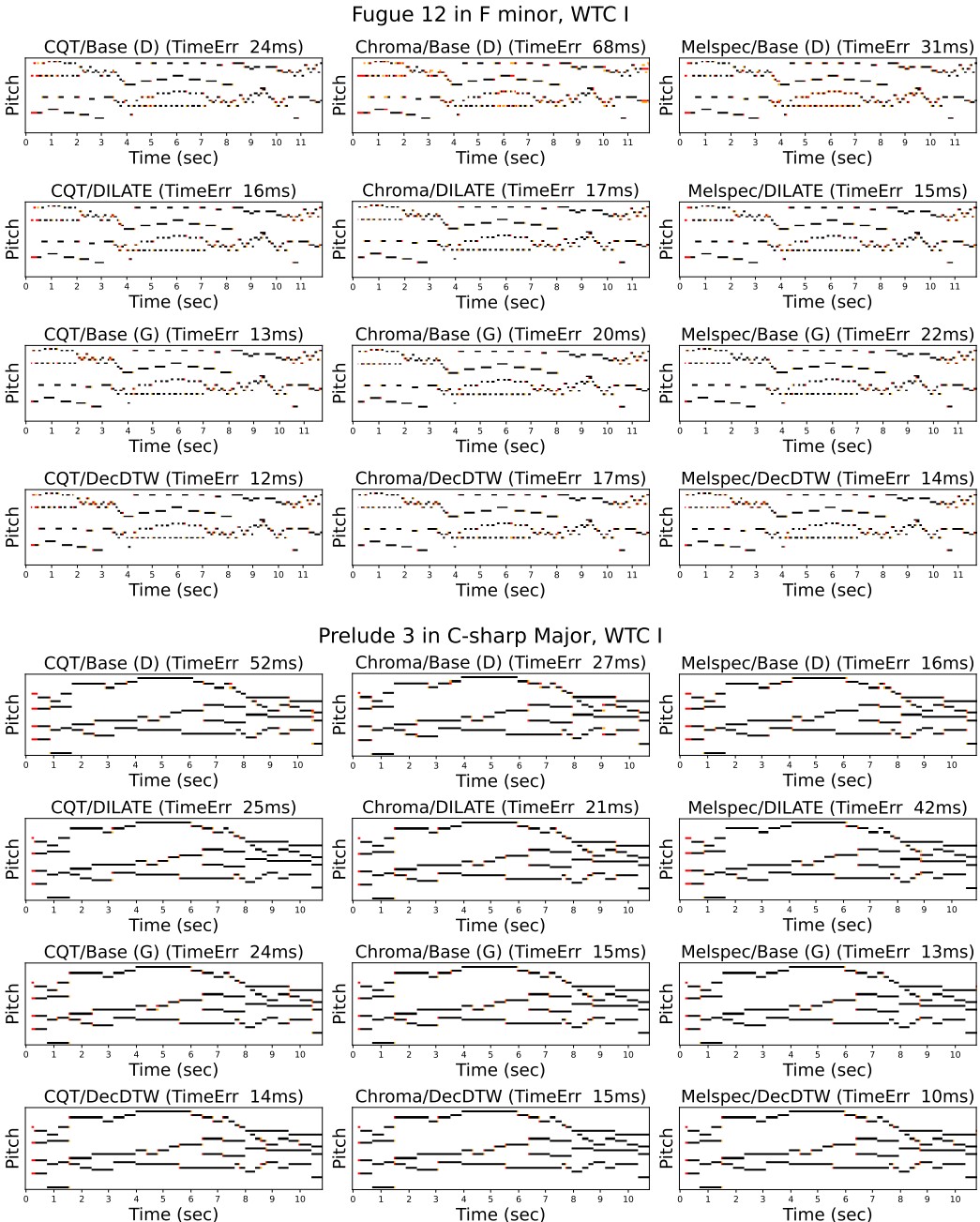

Figure 8: Example of four randomly selected test set alignments for the audio-to-score alignment experiments. All recordings are performances of segments from Bach's *The Well Tempered Clavier*. Each column represents a particular feature type and each row represents a particular training loss.

**Overcast Query Max Error: Base (G) 5.26m, DecDTW 2.22m, Base (D) 23.35m, DILATE 7.85m**

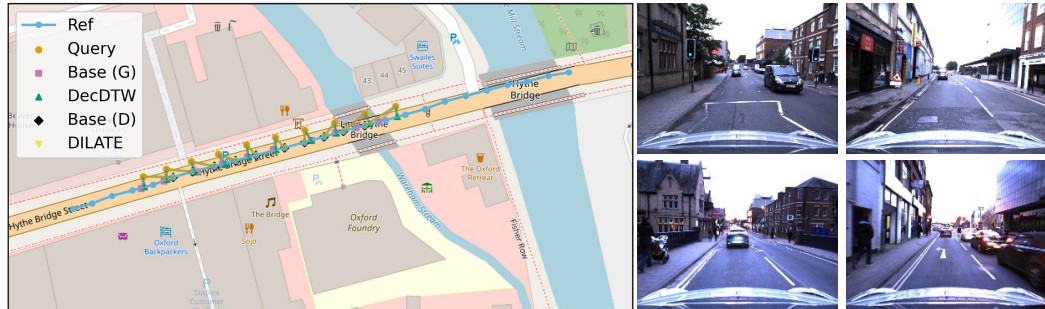

**Snow Query Max Error: Base (G) 5.77m, DecDTW 3.86m, Base (D) 10.40m, DILATE 6.14m**

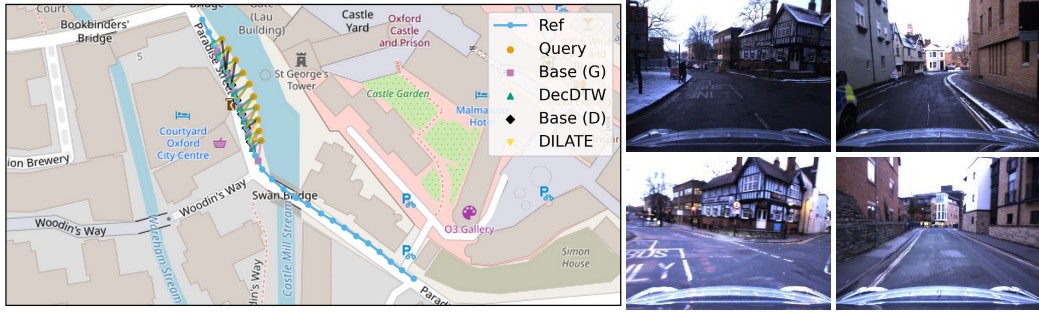

**Sunny Query Max Error: Base (G) 4.33m, DecDTW 3.52m, Base (D) 21.29m, DILATE 15.48m**

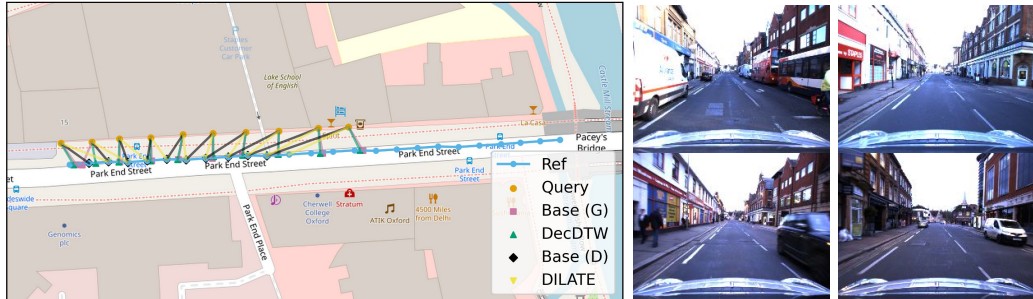

**Overcast Query Max Error: Base (G) 2.31m, DecDTW 7.56m, Base (D) 10.22m, DILATE 4.66m**

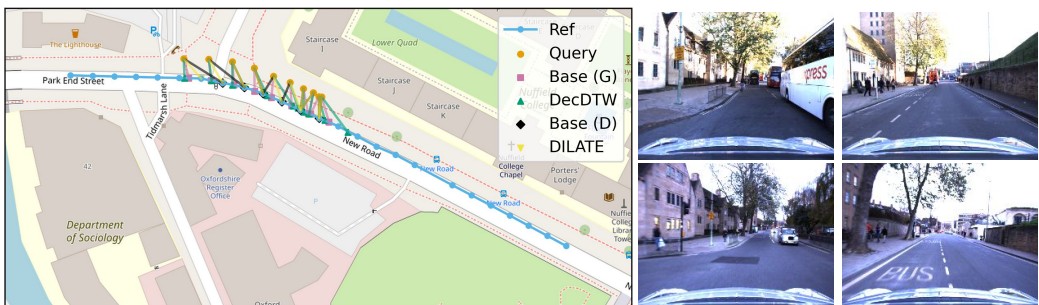

Figure 9: Additional example localisation output for the VPR experiments selected from the test set. Example query images from the sequence are provided in the top row and example database images are provided in the bottom row. Query poses are offset from the database for visualisation purposes only. The substantial intra-sequence velocity change in the last (bottom) example, combined with warp regularisation causes DecDTW training to perform worse than Base (G) and DILATE.

