# OpenReview forum: "Deep Declarative Dynamic Time Warping for End-to-End Learning of Alignment Paths"
_ICLR.cc/2023/Conference — ICLR 2023 poster_

### Official Review · Reviewer_yA9i · 2022-10-14

**Confidence:** 4
**Correctness:** 4
**Technical Novelty And Significance:** 3
**Empirical Novelty And Significance:** 2
**Recommendation:** 6

**Clarity, Quality, Novelty And Reproducibility:**

The description of the method is rather clear.
The idea to use the Generalized DTW formulation in order to differentiate through the resulting path is novel, but an important related work is missing.
In terms of reproducibility, for reasons I listed above, I would have a hard time reimplementing the experiments and I believe their presentation should be re-worked.

**Strength And Weaknesses:**

* Strengths
    * Using the path information from DTW alignments in gradient-based optimization is an interesting track of research
    * Relying on the Generalized DTW formulation to do so makes a lot of sense
* Weaknesses
    * The major weakness in this paper is that it misses an important related work [1] that also relies on integrating the path information (from SoftDTW this time) for gradient-based optimization with a DTW-derived loss.

        1. Shape and Time Distortion Loss for Training Deep Time Series Forecasting Models. Le Guen & Thome, NeurIPS 2019

    * Another weakness of the paper is that the presentation of the experiments is too brief, which makes it hard to get what each competing method does in practice.

In more details:

* At several places, authors suggest that when using SoftDTW, one cannot differentiate through the path information, which is not true (it is discussed in the original SoftDTW paper and done in [1]). Below are a few occurrences of this claim in the paper:
    > DTWNet and SoftDTW only allow the DTW discrepancy between two time series to be differentiable wrt its inputs (Legend of Fig 1)

    > This enables for the first time the use of the alignment path itself in a downstream loss (page 2)

    * Comparison to [1] is of prime importance given that the computation for DecDTW is said to be "between 4-20 times slower than SoftDTW"
* The description of the experiments is sometimes hard to follow. As an example, I do not understand Table 1. More specifically, I do not understand what the subparts (left and right) of the Table are, since we have a first subpart dedicated to "Classic DTW" in which there is still a DecDTW entry and similarly a second subpart named "GDTW" in which there are DTWNet and SoftDTW entries.
I guess there is something like one can use a different alignment algorithm to train the network and predict a path, but if this is the explanation, things should be made much clearer in the text in my opinion.
* Here are other remarks related to the experiments:
    > Interestingly, fine-tuning using DecDTW dramatically reduces independent retrieval performance (p. 9)
       * This would be worth a discussion
    * Also, I do not get why "DTW (Base)", "Fine-tune Along GT" and "Fine-tune OTAM" reach the exact same level of performance here. Is there a rational explanation for this fact?
* Finally, this is a minor remark, but I feel that, when presenting GDTW, authors should not use sentences such as:
    > This more general description of the DTW problem

    > The dynamic time warping problem can be formulated as follows (page 3)

    since the GDTW problem is not the DTW problem (yet it is an interesting formulation anyway)

**Summary Of The Paper:**

This paper suggest the use of Generalized Dynamic Time Warping as a way to get differentiable alignments.
It then showcases the utility of this method on experiments on which the loss to be optimized depends directly on the obtained path.

**Summary Of The Review:**

Overall, this paper introduces interesting ideas, but the lack of comparison to an important baseline (SoftDTW with regularization on the path [1]) makes it impossible to assess whether the proposed approach outperforms the said baseline.

---

> ### Author Response · Authors · 2022-11-15
> **Thank you for your review [1/2]**
>
> Thank you for your review. We believe we have addressed your main concerns, i.e. included the provided comparison method to the experiments, and have improved the explanation of the experiments as per your (and the other reviewers’) feedback. In addition, we have changed the commentary throughout the text to remove any claims of being the first to incorporate path information in a differentiable way. We hope you will consider modifying your score during the discussion period to reflect these changes.
>
> > The major weakness in this paper is that it misses an important related work [1] that also relies on integrating the path information (from SoftDTW this time) for gradient-based optimization with a DTW-derived loss.
> > 1. Shape and Time Distortion Loss for Training Deep Time Series Forecasting Models. Le Guen & Thome, NeurIPS 2019
>
> We thank the reviewer for pointing out the missed comparison to [1]. Admittedly, this was our mistake and we missed this work in our literature review. We have added a comparison to DILATE using the reference implementation provided by the authors of that paper.  While DILATE improves upon base features in both experiments, we find that DecDTW is still the most effective at utilising path information by a large margin. Partially, this is due to GDTW itself; warp regularisation and the ability for alignments to interpolate between observed times naturally improves alignment accuracy, even for baseline features. However, our experiments show that DecDTW yields more substantial gains compared to the base GDTW after learning (versus DILATE over classic DTW). See Tables 1 and 2 for the updated results and Sections 7.1 and 7.2 for the updated commentary around the comparison methods (including DILATE) and discussion around the results.
>
> > Another weakness of the paper is that the presentation of the experiments is too brief, which makes it hard to get what each competing method does in practice.
>
> We apologise for the brevity of experiments in the existing manuscript. We have added more details in the supplementary material for the revised document around the specifics of the comparison methods to address the concerns presented by you and the other reviewers. This will cover the full pipeline, i.e. raw observations are passed through a deep network to recover a time series of learned features, which are then fed into a downstream loss (e.g. SoftDTW discrepancy, DILATE loss). After training using the loss and selecting the best model using the validation set on the end task metric (e.g. alignment error), we evaluate each method by producing alignments using the associated alignment technique. This is classic DTW for DILATE, SoftDTW, Along GT and GDTW for DecDTW.
>
> Compressing experiments for two real-world problems (with associated related works and problem formulation) into the space available in the main paper is challenging. To aid reproducibility, we have provided all of our code (including data generation and training code for experiments) in the original submission and hope this helps alleviate your concerns.
>
> > Comparison to [1] is of prime importance given that the computation for DecDTW is said to be "between 4-20 times slower than SoftDTW"
>
> In addition to comparing alignment results, we have also included DILATE in the scalability analysis (see our response to reviewer Z1je). Our analysis indicates that DILATE is in fact slower than DecDTW during train time. However, we do not deny that the ability to use classic DTW to produce alignments afforded by DILATE is still beneficial w.r.t. compute time.

---

> > ### Author Response · Authors · 2022-11-15
> > **Thank you for your review [2/2]**
> >
> >
> > > The description of the experiments is sometimes hard to follow. As an example, I do not understand Table 1. More specifically, I do not understand what the subparts (left and right) of the Table are, since we have a first subpart dedicated to "Classic DTW" in which there is still a DecDTW entry and similarly a second subpart named "GDTW" in which there are DTWNet and SoftDTW entries. I guess there is something like one can use a different alignment algorithm to train the network and predict a path, but if this is the explanation, things should be made much clearer in the text in my opinion.
> >
> > Yes, your interpretation is correct. We apologise that this was confusing in the text, and have changed the table structure significantly in the revised manuscript as a result. For both sets of experiments, we provide summary results in the main body and additional results in the supplementary.
> >
> > In the main text, we change the tables to couple the alignment method with the loss. For example in the new Table 1 and 2, methods trained using SoftDTW/OTAM, Along GT, DTWNet, DILATE uses classic DTW to produce alignments at test time. Methods trained using DecDTW uses GDTW to generate alignments at test time.
> >
> > We however keep our ablations for the audio experiments, where after training we apply different alignment methods (e.g., for a DecDTW trained model, we still evaluate producing DTW alignments from it). These ablations are now stored in the appendix. The conclusions of the results from the ablation do not differ from the original manuscript. Furthermore, we have added more details when explaining this ablation in the appendix.
> >
> > > Here are other remarks related to the experiments:
> > > Interestingly, fine-tuning using DecDTW dramatically reduces independent retrieval performance (p. 9) * This would be worth a discussion.
> >
> > This is indeed worth further discussion. The baseline features are trained to optimse single image retrieval performance using a contrastive loss over single images. However, the DecDTW/DILATE loss is designed to maximise performance accounting for GDTW/DTW alignment. It is interesting (especially for DecDTW) that significant gains in sequence alignment performance comes at a significant cost to single image retrieval performance. In the updated manuscript, we store these results with full discussion in the appendix.
> >
> > > Also, I do not get why "DTW (Base)", "Fine-tune Along GT" and "Fine-tune OTAM" reach the exact same level of performance here. Is there a rational explanation for this fact?
> >
> > This is a good question. While training, we use the alignment error over the validation set to select the model and evaluate this 5 times per epoch (and once before training starts). The validation criterion (max alignment error) immediately starts to increase for Along GT and OTAM when training, and so the best model is in fact, the model before any fine-tuning occurs. We have updated the revised manuscript in the main body to include this explanation.
> >
> > >  Finally, this is a minor remark, but I feel that, when presenting GDTW, authors should not use sentences such as:
> > > This more general description of the DTW problem
> > > The dynamic time warping problem can be formulated as follows (page 3)
> > > since the GDTW problem is not the DTW problem (yet it is an interesting formulation anyway)
> >
> > Here we were talking about the solver; i.e. classic DTW just constrains the discretisation to the grid. We acknowledge this is not the formulation itself, hence possibly confusing. We will make reference to GDTW as a completely separate (although) similar method to DTW and remove any claims of generalisation to DTW that you mentioned above.

---

> > > ### Comment · Reviewer_yA9i · 2022-11-17
> > > **Response to authors**
> > >
> > > I would like to start by thanking authors for this detailed and insightful discussion around the points I raised in my review.
> > >
> > > **First**, regarding the reference to DILATE.
> > > My main claim was that the mere existence of the DILATE paper made some of the claims in the paper irrelevant.
> > > The authors have fixed these occurrences and this is an important improvement of the paper in my opinion.
> > > However, when I was asking about comparing to DILATE, I meant positioning the proposition wrt DILATE.
> > > I am not sure direct experimental comparison to DILATE makes a lot of sense since the loss formulation in DILATE forces the alignment path around the diagonal, while the proposed DecDTW relies on the ground truth alignment, which is its strength.
> > > If authors wanted to compare the use of a ground truth alignment in DILATE versus the one in DecDTW, this would require to alter the DILATE method and have a Gaussian penalization around the ground truth path instead of a Gaussian penalization around the diagonal, as done in the original DILATE formulation.
> > > I guess this is not what is done in the presented experiments (which is OK: this would have been a lot of work for a rebuttal period I think).
> > > This is especially critical since, according to this text in the supplementary material:
> > >
> > > > For DILATE, we found that using $\alpha = 0$ (temporal loss only) and $\gamma = 1$ yielded the best results
> > >
> > > the DILATE method is not trained for softDTW loss but only for the part of the loss that pushes the obtained alignment towards the diagonal.
> > >
> > > **Second**, my other important concern was about the clarity of the experimental section, and I have to admit that authors have made a remarkable work on this matter: the experimental Section is much clearer now in my opinion.
> > >
> > >
> > > **Minor remark:**
> > >
> > > > DTW is used for alignments in Base (D), SoftDTW, Along GT and DILATE, while GDTW is used for alignments in DecDTW only.
> > >
> > > I believe that GDTW is used for both DecDTW and Base (G).
> > >
> > > **For all these reasons**, I decide to slightly improve my rating but still vote towards rejection because of the comparison to DILATE which requires more work.

---

> > > > ### Author Response · Authors · 2022-11-17
> > > > **Further clarification**
> > > >
> > > > Thank you for your response. We appreciate you acknowledging the improvements we have made within the revised manuscript and in revising your score accordingly. However, we would like to respond to your new comments:
> > > >
> > > > > However, when I was asking about comparing to DILATE, I meant positioning the proposition wrt DILATE.
> > > >
> > > > Thank you for clarifying this. I believe we can succinctly summarise the positioning of DecDTW w.r.t. DILATE by the following points:
> > > >
> > > > - DecDTW is a fundamentally different DTW algorithm (GDTW vs DTW). This has its own advantages, even before considering differentiability, as we show in the experiments (better alignments w/regularisation and continuous times).
> > > > - The temporal loss in DILATE attempts to utilise path information via a soft approximation to the full $m\times n$ path matrix. Our differentiable DTW formulation allows gradient computation directly on the exact path.
> > > > - Our direct comparison to DILATE yields two key observations: First, DecDTW appears much more effective at utilising path information (even after accounting for the GDTW -> DTW gap) during learning. Second, DecDTW actually affords *faster* training iterations versus DILATE.
> > > >
> > > > We have modified the revised manuscript to even further make this positioning clear. This reflects the changes we have already made to Section 1 (introduction) and Section 7 (experiments).
> > > >
> > > > > If authors wanted to compare the use of a ground truth alignment in DILATE versus the one in DecDTW, this would require to alter the DILATE method and have a Gaussian penalization around the ground truth path instead of a Gaussian penalization around the diagonal, as done in the original DILATE formulation. I guess this is not what is done in the presented experiments (which is OK: this would have been a lot of work for a rebuttal period I think). This is especially critical since, according to this text in the supplementary material:
> > > >
> > > > We agree that this detail is critical and will clarify specifically how we implemented DILATE. We did indeed modify the original source code to incorporate information from ground-truth alignments into the $\Omega$ penalty matrix. In our comparisons, the penalty terms in $\Omega$ penalise deviation from the ground-truth alignments, as opposed to the diagonal, as in the original paper. We believe as a result, this makes the comparison between DecDTW and DILATE to be a fair one. Furthermore, the error encoded into $\Omega$ is given by the *task* error, which is how we trained DecDTW. This is using an absolute value penalty based on the alignment error for audio (TimeErr) and a squared penalty based on GPS error itself (rather than temporal alignment error) for the vision task. This additional detail further makes the comparison more "like-for-like" (rather than just a Gaussian penalty around the ground-truth path).
> > > >
> > > > The details around how we implemented DILATE and parameterised $\Omega$ are already available in the appendix (see **Comparison Methods** paragraphs in A4 and A5. Furthermore, we have improved clarity in the main body in Section 7 and have added comments explicitly stating that we modified DILATE to use ground truth information.
> > > >
> > > > UPDATE: revised manuscript updated again. We hope the changes address your current concern around the DILATE comparisons.
> > > >
> > > > Another UPDATE: Updated this response and the manuscript to reflect the changes to positioning against DILATE in the text.

---

> > > > > ### Comment · Reviewer_yA9i · 2022-11-18
> > > > > **Response to response to... :)**
> > > > >
> > > > > Thqnk you, this was not clear for me when reading the Section 7 describing the use of DILATE.
> > > > > If ground truth alignments ar used for DILATE, then I further edit my vote to weak accept.
> > > > > Thank you for this fruitful rebuttal period.

---

> > > > > > ### Author Response · Authors · 2022-11-18
> > > > > > **Final response? :)**
> > > > > >
> > > > > > Thank you very much for your contributions to our paper!

---

### Official Review · Reviewer_Z1je · 2022-10-24

**Confidence:** 4
**Correctness:** 3
**Technical Novelty And Significance:** 3
**Empirical Novelty And Significance:** 3
**Recommendation:** 6

**Clarity, Quality, Novelty And Reproducibility:**

The paper is clearly written and novel to the community. Code and data are provided for reproduction.

**Strength And Weaknesses:**

Pros:
1. The idea is novel that is very different from existing differentiable DTW methods (e.g. DTWnet or soft-DTW). This is the first attempt of using declarative network in DTW problem.
2. The experiments are conducted on some interesting and very practical tasks, the audio-to-score and visual query.
3. The paper is clearly written and easy to follow. Additional details are provided in appendix.

Cons:
1. The motivation of applying declarative network is vague. If just for incorporating alignment path error, simple modifications can be done on DTWNet or SoftDTW to obtain the DTW path as a byproduct, and adding the path into the loss is achievable by defining a proper distance between hypothesis path and gold alignment.
2. Lack of comparison with other methods that regularize the alignment paths, e.g. Graphical Time Warping (GTW) and following literatures on this method.
3. The method is not scalable, 2 orders of magnitude slower than other methods. This significantly limit the usage of the method. This is due to the complexity in declarative network.

**Summary Of The Paper:**

This paper proposes a novel DTW layer that combines non-linear programing based generalized DTW (Deriso & Boyd) with declarative network (Gould et al.), to not only output the DTW discrepancy value, but also the alignment. More importantly, the output alignment could be compared against ground truth alignment and the difference could be backpropogated to train the network. Experiments on audio-to-score alignment and image query datasets, both show improvement over SOTA methods. Scalability is also discussed in the end.

**Summary Of The Review:**

Overall, the method is novel but lack some motivation behind it. The applications are interesting to the community.

---

> ### Author Response · Authors · 2022-11-15
> **Thank you for your review**
>
> Thank you for your review. We believe we have addressed your concerns around motivation and scalability in our detailed responses, and have incorporated the relevant changes in the revised manuscript.
>
> > The motivation of applying declarative network is vague. If just for incorporating alignment path error, simple modifications can be done on DTWNet or SoftDTW to obtain the DTW path as a byproduct, and adding the path into the loss is achievable by defining a proper distance between hypothesis path and gold alignment.
>
> We agree with your comment around alignment path error alone; Reviewer yA9i pointed to a method for training with ground truth alignments using SoftDTW. We still believe however, that our method is quite novel compared to existing methods since the optimal alignment path itself (as a length N vector) is differentiable. DILATE [1], in comparison, uses an approximation to the optimal $N\times M$ path matrix.
>
> In addition to being methodologically distinct to existing methods, we show in our revised experiments against DILATE that DecDTW is significantly more effective at utilising path information, especially for challenging real-world tasks. Finally, we observe that DecDTW is faster for training compared to DILATE. See our response to reviewer yA9i for tables containing revised experimental results and point 3 below for the scalability analysis.
>
> [1] Shape and Time Distortion Loss for Training Deep Time Series Forecasting Models. Le Guen & Thome, NeurIPS 2019
>
> > Lack of comparison with other methods that regularize the alignment paths, e.g. Graphical Time Warping (GTW) and following literatures on this method.
>
> Thanks for pointing us to GTW [B] and its following literature [C,D], which also employs a regularization hyperparameter. Since GTW’s source code is not publicly available, we have been unable to produce comparisons against it so far during this discussion period. Nevertheless, we have now included a citation to this line of work at the end of Section 3 of the revised manuscript, as follows:
>
> “Regularisation is crucial for preventing noisy and/or pathological warps  (Zhang et al., 2017; Wang et al., 2016) from being produced from GDTW (and DTW, more generally), and can greatly affect alignment accuracy. $\lambda$ can be selected optimally using cross-validation (Deriso and Boyd, 2019).”
>
> [B] Wang, Yizhi, et al. "Graphical time warping for joint alignment of multiple curves." Advances in Neural Information Processing Systems 29 (2016).
>
> [C] Wu, Chiung-Ting, et al. "Alignment of LC-MS Profiles by Neighbor-wise Compound-specific Graphical Time Warping with Misalignment Detection." bioRxiv (2019): 715334.
>
> [D] Strøm, Aleksander S. Relative Geologic Time By Dynamic Time Warping. MS thesis. uis, 2022.
>
> > The method is not scalable, 2 orders of magnitude slower than other methods. This significantly limit the usage of the method. This is due to the complexity in declarative network.
>
> To address your concerns around scalability, we revamped the entire scalability analysis section to provide more analysis and information. We split the analysis between train and test time scalability and briefly describe the conclusions as follows (see the revised manuscript for plots).
>
> During test time, we can trade-off warp fidelity with computation time directly for GDTW. By removing all rounds of iterative refinement (GDTW approx.), we can produce warps significantly faster with only a mean warp estimation error of 0.4% (6% max error). Full GDTW with iterative refinement is 15-50x slower than classic DTW, however GDTW with no iterative refinement is only 4-12x slower. For the train time scalability analysis, we observed that DILATE, which also utilises path information, is in fact slower than DecDTW. We believe the efficient, explicit computation of the backward pass is faster than the method proposed in DILATE.
>
> We also believe this computation time can be further greatly reduced in future work by using a different parameterisation of the warp function (e.g. polynomial basis). A more compact parameterisation may substantially improve the efficiency of both the forward and backward pass. Refer to the below paper for a rough idea of how this might look.
>
> [F] Feng Zhou and Fernando De la Torre. Generalized Time Warping for Multi-modal Alignment of Human Motion. CVPR 2012
>
> We still agree with the reviewer that scalability is currently a limitation of our method. However, we can see that DecDTW performs substantially better than comparable methods that utilise path information (see response to reviewer yA9i around DILATE comparison) and we believe this makes our method a valuable contribution to the community nonetheless.

---

### Official Review · Reviewer_EEJt · 2022-10-25

**Confidence:** 4
**Correctness:** 4
**Technical Novelty And Significance:** 3
**Empirical Novelty And Significance:** 3
**Recommendation:** 8

**Clarity, Quality, Novelty And Reproducibility:**

**Clarity**

This paper is refreshingly well-written overall. In spite of this, I found it a bit difficult to follow some important aspects of the core argument. Some examples / suggestions:

- I was searching early on for the connection between the proposed method and well-known algorithms like CTC which also learn alignments between time series pairs. It took me quite a while to understand the high-level distinctions between CTC and the proposed DecDTW (namely, that CTC jointly learns to align and transcribe and does not require a cost function, while DTW learns to align and requires a cost function). It would be helpful for readers to clarify these distinctions in the related work section.
- I was initially thinking that all of the methods being proposed / compared would output an _alignment_ rather than _features_. This wasn’t made clear to me until Table 6 (and even then it took me quite some time to understand). The source of my confusion was likely Figure 1 where “Feature extractor” is an _input_ to DecDTW; perhaps the authors could mention earlier on that the proposed methods can all act as sort of “secondary feature extractors”, and that these features can be compared using both classic DTW and GDTW (as in Table 6)? Additionally, the meaning of “features” in 3.1 Preliminaries could stand to be clarified.
- I was caught off guard by the strong performance of GDTW compared to Classic DTW on _all_ features in Table 6 - perhaps the authors could set this expectation (namely, that GDTW >> Classic DTW) earlier on in related work?
- Section 3.2 could remind readers that we’re talking about GDTW (I had forgotten after spending time scrutinizing the preliminaries. E.g., “GDTW formulates DTW as a constrained continuous optimisation problem…”
- In Section 4, could the authors clarify at a high level why we are now returning to a discretized formulation after spending all that effort instantiating a continuous formulation?

Low-level:
- (Figure 3) “can applied” -> “can be applied”
- (Section 4) “It is easy to see” pet peeve - this is _not_ easy for everyone to see (myself included)

**Quality**

I often find there is an unfortunate tradeoff in ML papers between the elegance / descriptive clarity of the methods and the application of those methods to real-world empirical settings. I was pleasantly surprised to find both in this paper. The experiments look at two very different but realistic alignment settings, and show strong performance in both cases.

**Novelty**

This paper draws heavily on Deriso & Boyd 2019, but has substantial novelty in its combination w/ Gould et al. 2021 to embed the GDTW framework into neural networks. One novel aspect w.r.t. other work in alignment is the ability to integrate ground-truth alignments during training. The actual utility of this is a bit unclear to me, e.g., in the case of music alignments, the ground-truth alignments are only available here due to unusual circumstances (the Disklavier yields alignments for MAESTRO and synthesis yields alignments for Kernscores). I would have loved to see the authors attempt to tackle the problem of jointly learning to align and transcribe, though I cannot fault them for putting this out of scope.

**Reproducibility**: In contrast to much of deep learning literature, it appears to be possible to reproduce this work from the information present in the paper alone. As an added bonus, the authors include code.


**Strength And Weaknesses:**

A primary strength of this paper is the combination of the theoretical elegance of the proposed method alongside its impressive performance on real-world alignment tasks. Additionally, this paper is well-written and contains precise notation. One weakness is that, while the proposed method is continuous and can be combined w/ neural networks, it does not address the problem of jointly learning to align and _transcribe_, unlike other methods like CTC which can learn alignments without designing an explicit cost function between input and output sequences (e.g., in speech recognition).

**Summary Of The Paper:**

**Update 11-16-22** I acknowledge the revisions made by the authors. I believe the proposed revisions further improve this already-strong submission. I am holding my score at 8 Accept and am willing to further champion this paper's acceptance if needed.

This paper proposes an algorithm for improving temporal alignments between related pairs of time series data, applicable in applications like audio-to-score alignment in music information retrieval. Unlike past efforts, the proposed algorithm can leverage ground truth alignment information during training if available, leading to improved performance on several real-world alignment tasks.

**Summary Of The Review:**

Overall, I think this paper presents a well-motivated method w/ strong empirical results on an important task. The scope of the exploration is somewhat limited to aligning (as opposed to _transcribing_ as in tasks like speech recognition), but this is an important task for which there are certainly useful real-world applications. Overall, I think the ICLR community will benefit from this paper’s inclusion and the ensuing discussion.

---

> ### Author Response · Authors · 2022-11-15
> **Thank you for your review [1/2]**
>
> Thank you for your review and your helpful suggestions around improving the clarity of the manuscript. We will add information around distinguishing our method from methods for transcription such as CTC in the related works in the revised manuscript. We will also address your editorial comments to improve the clarity of the writing in the revised manuscript.
>
> > A primary strength of this paper is the combination of the theoretical elegance of the proposed method alongside its impressive performance on real-world alignment tasks. Additionally, this paper is well-written and contains precise notation. One weakness is that, while the proposed method is continuous and can be combined w/ neural networks, it does not address the problem of jointly learning to align and transcribe, unlike other methods like CTC which can learn alignments without designing an explicit cost function between input and output sequences (e.g., in speech recognition).
>
> We appreciate your comments around theoretical elegance and real-world alignment tasks; this was certainly our intention when submitting the paper. We acknowledge that a limitation of this method is the focus on alignment with an explicit cost function and lack of automatic transcription (notably, in the audio experiments and less so in the VPR one). However, this limitation also applies to other related methods on differentiable DTW. In addition, for localisation tasks like the VPR and audio-to-score task, the explicit alignment path itself (rather than a transcription) is of interest. We hope that future work around cross-modal DTW-based alignment could be a viable first step towards using our method for the transcription task. We will address your specific comments around the limitations in more detail below.
>
> > I was searching early on for the connection between the proposed method and well-known algorithms like CTC which also learn alignments between time series pairs. It took me quite a while to understand the high-level distinctions between CTC and the proposed DecDTW (namely, that CTC jointly learns to align and transcribe and does not require a cost function, while DTW learns to align and requires a cost function). It would be helpful for readers to clarify these distinctions in the related work section.
>
> Thank you for bringing this to our attention and helping to improve the clarity of our paper. We have now included the following clarification in the revised manuscript:
>
> “We note that methods based on DTW differ to methods such as CTC for speech recognition [A] where word-level transcription is far more important than frame-level alignment, and an explicit alignment step is not required.”
>
> [A] Graves, Alex, and Navdeep Jaitly. "Towards end-to-end speech recognition with recurrent neural networks." International conference on machine learning. PMLR, 2014.
>
> > I was initially thinking that all of the methods being proposed / compared would output an alignment rather than features. This wasn’t made clear to me until Table 6 (and even then it took me quite some time to understand). The source of my confusion was likely Figure 1 where “Feature extractor” is an input to DecDTW; perhaps the authors could mention earlier on that the proposed methods can all act as sort of “secondary feature extractors”, and that these features can be compared using both classic DTW and GDTW (as in Table 6)? Additionally, the meaning of “features” in 3.1 Preliminaries could stand to be clarified.
>
> We have received similar comments from reviewer yA9i around presentation of comparison methods in the experiments and have revised the tables and commentary around this point. To specifically address your concern, we clarified the commentary in the experimental setup to reflect each method outputting an alignment. In practice, a method trained on SoftDTW would be coupled with classic DTW to generate alignments, whereas a method trained using a DecDTW loss would use GDTW for alignments.
>
> However, we included the ablation analysis present in the original paper (changing alignments for trained features for a given loss) as extra analysis in the appendix. We believe this “specialisation” of feature extractors to a particular alignment method to be an interesting result which likely deserves a mention. The nature of the ablation is described in more detail compared to the original manuscript. We will also clarify in Figure 1 and the text throughout the definition of “feature”. Discussion in the experimental setup will also be updated to reflect the new discussion.

---

> > ### Author Response · Authors · 2022-11-15
> > **Thank you for your review [2/2]**
> >
> > > I was caught off guard by the strong performance of GDTW compared to Classic DTW on all features in Table 6 - perhaps the authors could set this expectation (namely, that GDTW >> Classic DTW) earlier on in related work?
> >
> > We were similarly surprised by the significant improvement of GDTW over classic DTW. We agree that this should be discussed earlier on, and to reflect this, we have now included the following text in the revised manuscript (both in Section 3):
> >
> > “We found in our experiments that GDTW outperformed DTW generally for real-world alignment tasks. We attribute this to the former’s ability to align in between observations, thus enabling a more accurate alignment.“
> >
> > “Regularisation is crucial for preventing noisy and/or pathological warps  (Zhang et al., 2017; Wang et al., 2016) from being produced from GDTW (and DTW, more generally), and can greatly affect alignment accuracy. $\lambda$ can be selected optimally using cross-validation (Deriso and Boyd, 2019)”
> >
> > > Section 3.2 could remind readers that we’re talking about GDTW (I had forgotten after spending time scrutinizing the preliminaries. E.g., “GDTW formulates DTW as a constrained continuous optimisation problem…”
> >
> > We have made the distinction between GDTW and DTW clearer overall in the revised manuscript, including for this section.
> >
> > > In Section 4, could the authors clarify at a high level why we are now returning to a discretized formulation after spending all that effort instantiating a continuous formulation?
> >
> > Our intent for the original manuscript was that the general, continuous formulation described in Section 3 is a more elegant way to introduce and define the GDTW problem compared to Section 4. This way, the nitty gritty around numerical integration (objective function) and assumptions around the warp function (piecewise linearity) would not obscure the main idea and formulation. We hope this encourages future work to change the implementation of GDTW while respecting the more abstract formulation. However, we acknowledge the lack of bridging statements and context for Section 4 and will add discussion around moving from the discretised to the continuous formulation in the revised manuscript.
> >
> > > I often find there is an unfortunate tradeoff in ML papers between the elegance / descriptive clarity of the methods and the application of those methods to real-world empirical settings. I was pleasantly surprised to find both in this paper. The experiments look at two very different but realistic alignment settings, and show strong performance in both cases.
> >
> > Thank you for pointing this out, we were hoping these qualities would be appreciated by readers.
> >
> > > This paper draws heavily on Deriso & Boyd 2019, but has substantial novelty in its combination w/ Gould et al. 2021 to embed the GDTW framework into neural networks. One novel aspect w.r.t. other work in alignment is the ability to integrate ground-truth alignments during training. The actual utility of this is a bit unclear to me, e.g., in the case of music alignments, the ground-truth alignments are only available here due to unusual circumstances (the Disklavier yields alignments for MAESTRO and synthesis yields alignments for Kernscores). I would have loved to see the authors attempt to tackle the problem of jointly learning to align and transcribe, though I cannot fault them for putting this out of scope.
> >
> > The key differentiator between DecDTW and prior methods based on DTW is that the optimal alignment path itself is fully differentiable. This does not restrict the application to only integrating ground truth alignments (despite it being a more obvious application), and we expect that future work will unlock some other applications or novel loss functions that can use the full alignment path. We also agree that proposing a novel method which uses DecDTW to tackle the joint transcription and alignment task would be a promising direction in future, e.g., see forecasting experiments in the DILATE paper provided by reviewer yA9i. For the time being, however, we have found two meaningful tasks to evaluate on where alignment itself is the desired goal, and there are likely other applications which we have not addressed explicitly in the paper.
> >
> > > Reproducibility: In contrast to much of deep learning literature, it appears to be possible to reproduce this work from the information present in the paper alone. As an added bonus, the authors include code.
> >
> > Thank you for acknowledging this. We hope transparency in our experiments and code can help the community more easily build on our work.

---

### Official Review · Reviewer_UgzC · 2022-11-01

**Confidence:** 3
**Correctness:** 1
**Technical Novelty And Significance:** 3
**Empirical Novelty And Significance:** 2
**Recommendation:** 6

**Clarity, Quality, Novelty And Reproducibility:**

The paper is well written. The narrative is straightforward, and the tasks are described in detail, which would benefit readers who aren't from music information retrieval or robotics backgrounds.

The formulations and results are explained reasonably well.

The code is relatively easy to read and cross-check with the declarative DTW formulation. The experiments are structured. As a nitpick, the code could be organized better (e.g. by modularizing classes), but it's already more than enough for a research paper.


**Strength And Weaknesses:**

Strengths:

- Directly outputs the alignment path
- Tested on two completely different domains
- Typically better results on both tasks compared to existing DTW variants

Weaknesses:

- Not compared to anything else but DTW-variants
- The experimental setup, in particular, the train/validation/test divisions are not explained in detail. See the specific comment below.
- Nit: The figures aren't colour-blind friendly
- For camera ready, adding a companion with audio and video examples could be good.

**Summary Of The Paper:**

The paper proposes a novel differentiable dynamic time-warping algorithm. The approach is advantageous over existing variants as it outputs a learnable warping path between the time series representations.

The authors test their approach on two distinct tasks: audio-score alignment and visual place recognition, and achieve results comparable to or better than the existing state-of-the-art.

**Summary Of The Review:**

The authors improve over DTW, a commonly used tool for sequential alignment tasks. Their formulation directly outputs the alignment path, which could be further used to improve the alignment. The paper is well written with minor corrections or clarifications required.

Given the above, I would like to recommend this paper to be presented at the conference.

Below I will list some minor/specific comments:

- Problem formulation & methodology: The text reads a bit like Thickstun is the first person to formalise audio-score alignment. I understand the authors refer to the wording in the cited paper. On the other hand, they also cite much earlier work (e.g. Ewert et al.) which has made similar definitions. It might be good to re-word; e.g. stating the pepper adopts the framework described in Thickstun et al.
- I think the setup needs more elaboration for both experiments. In both experiments, the authors use a subset of the data available, but they do not explain why. Moreover, in the audio-score alignment experiments, it is not explained if the scores could be present in multiple training/validation/test sets, potentially causing leakage.
- A7: It would also be good to include worse alignment results and discuss why such failures occur.
- Future work: it would be very interesting to extend the work on "subsequence alignment with jumps," see JumpDTW.

---

> ### Author Response · Authors · 2022-11-15
> **Thank you for your review**
>
> Thank you for your review as well as the additional feedback/suggestions. We appreciate that you enjoyed the writing overall and that you looked through our code. We believe we have addressed most of your concerns in our detailed responses below and will make the related changes to the revised manuscript. Please let us know if the figures in our revised manuscript are still not color-blind friendly (we used a specialised color template this time).
>
> > Problem formulation & methodology: The text reads a bit like Thickstun is the first person to formalise audio-score alignment. I understand the authors refer to the wording in the cited paper. On the other hand, they also cite much earlier work (e.g. Ewert et al.) which has made similar definitions. It might be good to re-word; e.g. stating the pepper adopts the framework described in Thickstun et al.
>
> We have made this change in the revised manuscript, making it clearer that we follow the Thickstun formulation rather than suggest anything about them being first.
>
> > I think the setup needs more elaboration for both experiments. In both experiments, the authors use a subset of the data available, but they do not explain why. Moreover, in the audio-score alignment experiments, it is not explained if the scores could be present in multiple training/validation/test sets, potentially causing leakage.
>
> Thank you for pointing out the potential score leakage issue. After re-examining our code, we can confirm that leakage does indeed occur for our audio experiments, where the same score is found across all splits. Performances, however, were not shared across splits. We agree with the reviewer that a better experiment would be to avoid leakage. To that end we have rerun our experiments with leakage removed. Even without score leakage, there are no changes to the conclusions of the experiments, despite minor changes to the numbers themselves. This further supports our claims. See our revised manuscript for the actual tables.
>
> In addition, we apologise for brevity in describing our experimental setup. We will describe the procedure for producing bootstrapped sequences from the 1M images for the VPR experiment in detail in the Appendix in the revised manuscript. For convenience, we will also describe the dataset construction procedure in the response. The objective of the paired sequence dataset is to have a sufficiently large and diverse train/val/test set. This is to ensure a network trained on the training set does not overfit, and moreover that evaluation is challenging and comprehensive. Sequence pairs are sampled from the 1M images to ensure both a geographically complete coverage of the areas covered by the RobotCar dataset for all splits. In addition, we wanted a diverse set of appearance condition pairs between query/reference. Reference sequences were generated for each reference traverse (we used 6 distinct dates spread across the 2 year period for training), with starting points separated by uniform spacing. For each reference sequence, we generated 24 query sequences randomly from the remaining pool of 50+ traverses.
>
> A large part of the reason why only a subset of the images are used is due to sparsity in the sequences. The single image dataset provided by Thoma et al., has images sampled at around 20Hz, whereas ours are sampled at 1Hz for query and 5m spacing for reference (see the original manuscript for a discussion around the selection of these) to reduce redundancy within image sequences. While we could certainly increase the dataset to use (almost) all of the images, we found this was not necessary to substantially improve alignment performance after training with DecDTW.  The specific sequence pairs used and splits are provided in the supplementary materials as a csv file, and we included the code used to generate the pairs as part of the submitted codebase.
>
> > A7: It would also be good to include worse alignment results and discuss why such failures occur.
>
> Noted, we will find failure cases and include them (with explanation) in the revised manuscript. We will update our response to this as soon as we have completed the required work around this.
>
> > Future work: it would be very interesting to extend the work on "subsequence alignment with jumps," see JumpDTW.
>
> We have included a sentence around integrating more sophisticated DTW formulations such as jumpDTW with DecDTW in the future works section of the revised manuscript.

---

> > ### Comment · Reviewer_UgzC · 2022-11-15
> > **Response to the authors**
> >
> > Thank you for addressing my comments,
> >
> > I am not sure if I have access to the changes, but I trust you to have carried them out.
> >
> > Re: re-rendering the Figures, thank you for making them colour blind friendly. As a nitpick, if you haven't done already, you could change the marker shape and the size for different elements in the legends, e.g. `x` has a dot marker and `y` has a diamond marker in Figure 2a, Figure 3 and so forth. Then your manuscript will be black-white friendly as well.
> >
> > It was a pleasure to read the submission.
> >
> > Best

---

> > > ### Author Response · Authors · 2022-11-16
> > > **Manuscript is updated**
> > >
> > > Thank you for the tips around figures. We have uploaded the manuscript now, and have addressed these new comments around markers in this updated version.

---

### Author Response · Authors · 2022-11-15
**Summary of response/changes**

Dear reviewers,

Thank you for taking an interest in our work, and for providing both feedback and helpful suggestions. We appreciated that reviewers were quite positive overall around the writing, novelty of our approach and the tasks used for evaluation. However, we acknowledged the missed comparison to a related piece of work, as well as a lack of clarity around both experimental setup and presentation of some of the results. We believe by addressing these comments during the discussion period, we were able to improve both the clarity of the writing and refine the claims and contributions w.r.t. existing works.

We believe we have addressed the majority of your concerns and have uploaded the revised manuscript. We look forward to further discussion during the discussion period, and will summarise the key changes in the revised manuscript below:

- We better positioned DecDTW compared to existing methods (e.g. DILATE [1]) which can utilise path information by adding to our literature review. This includes both discussing conclusions from the experiments (see third point), as well as providing a discussion around fundamental differences and novelties in methodology.
- We also removed discussions where we claim to be the first to incorporate path information, and instead focus our claims on being more effective and efficient at utilising path information. DecDTW allows gradient computation over the *exact* path, rather than the soft approximation afforded by DILATE. We also emphasise the inherent novelty around incorporating GDTW (which seems to show benefits for alignment tasks versus DTW, even before learning) in a differentiable way into deep networks.
- We added a comparison to DILATE [1] for both experiments, which incorporates differentiable path information with SoftDTW. We ensured this comparison was like-for-like by modifying the original DILATE temporal loss to account for ground truth alignments. Our conclusions are that while test time inference speed for producing alignments is still substantially faster for DILATE (due to using classic DTW for alignment), the alignment performance (and relative performance gain over the baseline) is still not nearly as strong training with DILATE compared to DecDTW. In addition, we found that DILATE is slower than DecDTW during training.
- Clarity of description for comparison methods and dataset setup have been improved both in the main body and the appendix.
- We have expanded on our original scalability analysis from the original paper. We split scalability analysis into train time and test time analysis and perform in-depth analysis for both cases.
- We have included additional relevant references and discussion requested by reviewers into the paper.

[1] Shape and Time Distortion Loss for Training Deep Time Series Forecasting Models. Le Guen & Thome, NeurIPS 2019

UPDATE: Revised manuscript has been uploaded. Note, qualitative example figures with additional comparisons to DILATE have not been updated (i.e. Fig 4 and A8, A9). This is due to issues with our computing cluster which is out of our control. We promise to update the figures for the camera-ready version to reflect the updated experiments if the paper is accepted.

---

### Comment · Area_Chair_XdZi · 2022-12-12
**Incorrect claims**

The paper makes a few incorrect claims, that need to be fixed.

> However, the classic DTW algorithm solves a discrete optimisation problem;

DTW can be cast as a linear program, so it's continuous optimization

> thus the DTW similarity is not differentiable w.r.t. the cost matrix.

This is wrong. The DTW value is continuous everywhere and is differentiable almost everwhere.
The DTW path is continuous almost everywhere and is piecewise constant (its Jacobian is therefore piecewise null).

> This is caused
by the use of min functions found in the DP recursion, preventing gradients
from being evaluated by unrolling the computations.

People backprop through relus. Why would min be problematic?

> use a continuous relaxation of min

Min is already continuous. What you want to say is differentiable-everywhere relaxation.

> which is not easily recoverable from the aforementioned approaches

For some reasons, the authors seem to think it's not possible to
differentiate through the soft-alignment produced by soft-DTW.
This is not true. Since the soft-alignment is the gradient of soft-DTW,
differentiating through it only requires Hessian products.
This is explained e.g. in https://arxiv.org/abs/2010.08354

> DecDTW outputs the exact, continuous time warp, which is differentiable

This contradicts the previous paragraph which says that the problem in Eq. (4) is nonconvex
and that approximate DP is used.

(7) and (8) only hold at the optimum, this should be explained

I suspect the differentiability is only true at the exact optimum of Eq. (4). If we only obtain an approximation, differentiability may no longer hold.

---

> ### Author Response · Authors · 2022-12-12
> **Response to the AC**
>
> Thank you for your comments. We appreciate the AC picking up some inaccuracies in the text. We will amend all of these points as discussed in the manuscript. However, we believe that the core contribution of a novel differentiable DTW layer (which distinguishes itself from SoftDTW and variants) stands, and that our experiments show that using DecDTW on real-world alignment tasks significantly outperforms using soft alignments.
>
> We provide a detailed response below:
>
> > DTW can be cast as a linear program, so it's continuous optimization
>
> This is a good point, we missed this point in the original submission. We will remove this claim of DTW being a discrete problem in the manuscript.
>
> > This is wrong. The DTW value is continuous everywhere and is differentiable almost everwhere. The DTW path is continuous almost everywhere and is piecewise constant (its Jacobian is therefore piecewise null).
>
> We agree regarding the DTW similarity being differentiable a.e. and will clarify this for the readers. It is possible that we will remove this claim (and the above one) altogether.
>
> > People backprop through relus. Why would min be problematic?
>
> This is a fair point. We will remove this statement in the manuscript.
>
> > use a continuous relaxation of min
>
> Agreed, we will change this in the text. Soft-DTW makes DTW differentiable, rather than differentiable almost everywhere. In addition, soft-DTW is claimed (in the original paper) to be much more stable compared to unrolling DTW itself due to the smoothing.
>
> Context: Regarding all of the previous comments, this was to discuss the history of differentiable DTW methods before SoftDTW. However, on reflection, there are mistakes in these claims, and we will revise them all in the manuscript. We will in general, revise these claims to remove false statements and be more consistent with those made in prior work (like the SoftDTW paper).
>
> > For some reasons, the authors seem to think it's not possible to differentiate through the soft-alignment produced by soft-DTW. This is not true. Since the soft-alignment is the gradient of soft-DTW, differentiating through it only requires Hessian products. This is explained e.g. in https://arxiv.org/abs/2010.08354
>
> This is exactly the method we compared against as suggested by reviewer yA9i (DILATE). We found that training using DecDTW yielded substantially better results over using the soft alignment path method when utilising ground-truth path information. In addition, DecDTW is verified in our scalability section (A.3) to be faster during training. We will change the wording to better reflect the availability of differentiable path information and reference the provided paper.
>
> > This contradicts the previous paragraph which says that the problem in Eq. (4) is nonconvex and that approximate DP is used.
> >
> > (7) and (8) only hold at the optimum, this should be explained
> >
> > I suspect the differentiability is only true at the exact optimum of Eq. (4). If we only obtain an approximation, differentiability may no longer hold.
>
> All of these points are correct. We added the "exact" in to try and differentiate our method from using soft-alignments. In practice, across a range of datasets, we found that applying an NLP solver after the DP (to better find the optimum) yielded no meaningful difference to the dynamics of learning. This is of course, subject to setting the discretisation and refinement in the approximate DP to be sufficiently high (e.g., discretisation equal to length of time series $M=n$ and 2-3 rounds of refinement). Insufficient discretisation or proximity to the optimum does indeed compromise the gradients and learning.
>
> We will make our claims more precise in the main text, to clarify to readers the effect of the solver and the approximation on the gradients in the backward pass. We will clarify our meaning of "exact" (single, optimal warp instead of a soft alignment). In addition, we will add some rules of thumb for setting DP solver parameters in A.1 in the manuscript.

---

### Decision · Program_Chairs · 2023-01-20

**Decision:**

Accept: poster

**Justification For Why Not Higher Score:**

The paper lacks rigour sometimes.

**Justification For Why Not Lower Score:**

Interesting idea and practical experiments.

**Metareview: Summary, Strengths And Weaknesses:**

The paper proposes a new DTW layer that combines a continuous formulation of DTW (Deriso & Boyd) with declarative networks (Gould et al.), i.e., implicit differentiation.

Strengths:
- While there are existing differentiable DTW approaches (e.g. DTWnet or soft-DTW), the proposed idea seems new.
- The experiments are conducted on some interesting and very practical tasks, e.g., audio-to-score and visual query.
- The paper is easy to follow.

Weaknesses:
- Several claims regarding DTW and soft-DTW are mathematically incorrect but the authors agreed to fix them.
- The paper sometimes lacks rigour regarding the proposed method, e.g., what happens when we discretize the continuous problem or when we only solve approximately the nonconvex problem.

The reviewers were enthusiastic about the paper. For the above reasons, I recommend acceptance as a poster.

**Note From Pc:**

if the above contains the word "oral" or "spotlight" please see: "oral" presentation means -> notable-top-5% and "spotlight" means -> notable-top-25%. As stated in our emails, we are disassociating presentation type from AC recommendations